# *VidEgoThink*: Assessing Egocentric Video Understanding Capabilities for Embodied AI

## Abstract

Recent advancements in Multi-modal Large Language Models (MLLMs) have opened new avenues for applications in Embodied AI. Building on previous work, EgoThink, we introduce VidEgoThink, a comprehensive benchmark for evaluating egocentric video understanding capabilities. To bridge the gap between MLLMs and low-level control in Embodied AI, we design four key interrelated tasks: *video question-answering*, *hierarchy planning*, *visual grounding* and *reward modeling*. To minimize manual annotation costs, we develop an automatic data generation pipeline based on the Ego4D dataset, leveraging the prior knowledge and multimodal capabilities of GPT-4o. Three human annotators then filter the generated data to ensure diversity and quality, resulting in the VidEgoThink benchmark. We conduct extensive experiments with three types of models: API-based MLLMs, open-source image-based MLLMs, and open-source video-based MLLMs. Experimental results indicate that all MLLMs, including GPT-4o, perform poorly across all tasks related to egocentric video understanding. These findings suggest that foundation models still require significant advancements to be effectively applied to first-person scenarios in Embodied AI. In conclusion, VidEgoThink reflects a research trend towards employing MLLMs for egocentric vision, akin to human capabilities, enabling active observation and interaction in the complex real-world environments.

## 1 Introduction

In recent years, Multi-modal Large Language Models (MLLMs; Du et al., 2022; Gan et al., 2022; Tang et al., 2023) have made significant strides in conventional vision-language tasks (Alayrac et al., 2022; Driess et al., 2023; Li et al., 2023b), profoundly impacting the field of Embodied Artificial Intelligence (Embodied AI; Ahn et al., 2022; Kuo et al., 2022; Huang et al., 2023; Zitkovich et al., 2023). Training data (Sharma et al., 2018; Schuhmann et al., 2022; Lin et al., 2014; Jia et al., 2021) for predominate MLLMs are typically collected from object-centric and exocentric perspectives, mirroring the distribution of conventional vision-language benchmarks (Liu et al., 2023; Xu et al., 2023; Li et al., 2023a; Ning et al., 2023), which focus primarily on object and scene understanding. However, to be effectively applied in Embodied AI, it is crucial not only to understand the surrounding environment but also to have extensive knowledge about the relationship between "myself" and the environment. For example, compared to the absolute position in the whole environment (e.g., "*the microwave is in the kitchen*"), the relative position to my body is more important (e.g., "*the microwave is one meter to my right*") for interaction and manipulation. Therefore, **egocentric videos** (Grauman et al., 2022; Damen et al., 2018), containing observations typical of third-person perspectives and additional interactions with the surrounding environment, can improve predominate MLLMs to be more general and expand their applications to the real world.

Various egocentric benchmarks (Cheng et al., 2024; Fan, 2019) have emerged to evaluate the capabilities of MLLMs from a first-person perspective. For instance, EgoTaskQA (Jia et al., 2022) and EgoPlan (Chen et al., 2023c) assess the planning capabilities of MLLMs for long-horizon tasks, while EgoSchema (Mangalam et al., 2024) aims to diagnose the understanding of very long-form video. However, the absence of a comprehensive video benchmark from the egocentric perspective presents a significant challenge to the development of general foundation models. Furthermore, current benchmarks, both in task design and textual output forms, focus on traditional video question-answering settings and neglect the potential to support downstream applications in Embodied AI,

# 🏋 VidEgoThink

Figure 1: The main tasks of VidEgoThink benchmark to comprehensively assess the egocentric video understanding capabilities in Embodied AI. There are four types of tasks, including *video question answering*, *hierarchy planning*, *visual grounding*, and *reward modeling*. These four tasks are complementary to each other to implement a complete goal for Embodied AI.

such as glass devices or autonomous robots. For example, the natural language output format (e.g., "*put salmon in microwave*") cannot be directly processed by robotics to take actions, whereas bounding boxes of grounded objects (e.g., "*microwave* [*290, 202, 835, 851*]" or function calls for low-level actions (e.g., "find(*microwave*)") align more closely with the input requirements of robotic control systems. Therefore, it is crucial to design suitable task formats that can be effectively applied to downstream applications in Embodied AI.

In this paper, we introduce *VidEgoThink*, as illustrated in Fig. 1, a comprehensive egocentric video understanding benchmark aimed at better aligning the capabilities of MLLMs for application in Embodied AI. Due to the stratospheric demand for training data of end-to-end Vision-Language-Action models (Driess et al., 2023; Padalkar et al., 2023; Li et al., 2024a), systems in Embodied AI are always structured into specialized hierarchical components. In detail, MLLMs can perform several key functions: (1) *video question-answering*, the basic module to comprehend the surrounding environment and human activities, and then generate corresponding responses to specific instructions (Cheng et al., 2024; Fan, 2019; Jia et al., 2022); (2) *hierarchy planning*, the core component to decompose high-level instructions to mid-level sub-goals and low-level actions (Ahn et al., 2022; Huang et al., 2022b;a); (3) *visual grounding*, the detector module to help Embodied AI system ground complex instruction to the physical world (Gao et al., 2023a; Chiang et al., 2024; Munasinghe et al., 2023); (4) *reward modeling*, the auxiliary module to classify task completion and further provide feedback according to the observations (Kwon et al., 2023; Di Palo et al., 2023; Yu et al., 2023). Rather than solely considering traditional question-answering or planning tasks like previous egocentric benchmarks, we specifically design these four tasks to comprehensively evaluate the capabilities for different functions of MLLMs in Embodied AI.

Considering the high cost of manually labeling data for four different tasks, we design a series of automatic construction pipelines leveraging existing annotations from the Ego4D dataset (Grauman et al., 2022). we use GPT-4o, known for its superior reasoning capabilities, to generate appropriate question-answering pairs by combining our designed prompts with existing human annotations. For the reward modeling task, we further adopt clipped images from each video to generate feedback for negative instances. To ensure diversity and quality, three annotators are asked to filter the automatically generated instances. For evaluation, we extensively compare 14 MLLMs across three categories: API-based MLLMs, open-source image-based MLLMs, and open-source video-based MLLMs. Experimental results indicate that all MLLMs perform poorly across all tasks. For example, GPT-4o with 32 frames and 8 frames achieve only 31.17 and 32.83 accuracy in video question-answering tasks. Detailed scores reveal that while MLLMs can determine existence across object, action, and scene dimensions, they particularly lack the ability to judge order or sequence. In other tasks, although GPT-4o's performance is subpar, other open-source MLLMs are almost completely unusable, showing significant performance gaps. Overall, applying current MLLMs directly to first-person scenarios in Embodied AI remains challenging and requires further effort. However, MLLMs hold great potential for advancing Egocentric Vision and Embodied AI, offering ample room for exploration and improvement.

## 2  RELATED WORK

**Multi-modal Large Language Models.** The advancement of large language models (LLMs; Brown et al., 2020; Ouyang et al., 2022; Wang et al., 2024) now extend into MLLMs. Visual modules, such as CLIP (Radford et al., 2021) and Q-Former (Dai et al., 2024), are integrated with pre-trained LLMs using various transition layers, equipping them with visual capabilities. From the wide selection of open-source LLMs, numerous image-based MLLMs (Chen et al., 2023b; Liu et al., 2024b; Zhang et al., 2023; Dai et al., 2024; Alayrac et al., 2022) have emerged. Moreover, the popularization of these image-based MLLMs has driven advancements in video perception. Video-based models like Video-LLaVA (Lin et al., 2023), Vision-LLaMA (Chu et al., 2024), and PandaGPT(Su et al., 2023) are capable of capturing the temporal information present in video form. In this work, we explore egocentric video understanding capabilities of MLLMs.

**Video-Langugae Benchmarks.** Numerous video-language benchmarks assess MLLMs, primarily focusing on instruction-following via visual question-answering tasks (Ning et al., 2023; Li et al., 2023d; Patraucean et al., 2023). Few benchmarks explore egocentric videos (Mangalam et al., 2024; Jia et al., 2022), like EgoTaskQA (Jia et al., 2022), EgoPlan-Bench (Chen et al., 2023c), and EgoGoalStep (Song et al., 2023). However, they often lack variety in assessed capabilities. Ego-Think (Cheng et al., 2024) covers more comprehensive capabilities but uses static images. Moreover, all these egocentric benchmarks with only conventional VQA tasks neglect that the designed task format should be grounded in the potential applications. Therefore, in this paper, we focus on comprehensively exploring the capabilities for different functions of MLLMs in Embodied AI. A comparison to recent video-language benchmarks is presented in Table 3 in Appendix A.

**Egocentric Video Datasets.** Egocentric video datasets (Grauman et al., 2022; Damen et al., 2018; Pirsiavash & Ramanan, 2012; Sigurdsson et al., 2018) capture first-person interactions with environment, aiding robotic tasks and augmented reality. These datasets are often recorded via head-mounted cameras or wearable glasses. As more egocentric videos become available, specialized datasets focusing on specific aspects of ego-perspective have emerged. For instance, LEMMA (Jia et al., 2020) includes data on goal-directed actions and multi-task situations. Ego-ExoLearn (Huang et al., 2024) and Ego-Exo4D (Grauman et al., 2024) emphasize egocentric videos that demonstrate an individual's understanding of activities when given an exocentric demonstration. These datasets provide a robust foundation for training and evaluating MLLMs from a first-person perspective.

## 3  TASK TYPES IN *VidEgoThink*

Given that the use of MLLMs in Embodied AI remains an open research question, we design four interrelated tasks, as shown in Fig. 1: video question-answering, hierarchy planning, visual grounding, reward modeling. The detailed descriptions of these four tasks are as follows.

### 3.1  VIDEO QUESTION ANSWERING

Previous evaluation studies on egocentric vision (Cheng et al., 2024) focus on static images, constrained by the input format limitations of earlier MLLMs. However, recent advancements in MLLMs (Achiam et al., 2023; Anthropic, 2024; Reid et al., 2024; Li et al., 2023c; Lin et al., 2023) have demonstrated significant progress. Since our real world is inherently dynamic, it is crucial to evaluate the video understanding capabilities of MLLMs.

**Dimensions.** To underscore the differences between static images and dynamic videos (Li et al., 2023d), we ensure questions require the entire video for accurate answers rather than just a single frame. We decompose video content around "myself" into three main elements: object, action, and scene. Furthermore, we explore fine-grained dimensions for each element, as shown in Fig. 2.

- **Object.** Egocentric videos emphasize the objects seen or used by "myself". We divide the object category into six dimensions: (1) *Object Existence (OE)*: Determining whether an object appears; (2) *Object Order (OO)*: Identifying the sequence of appeared objects; (3) *Object Interaction (OI)*: Assessing whether and how an object has been used; (4) *Object Count (OC)*: Counting the total number of objects for a specific type; (5) *Object State (OS)*: Assessing whether the state of an object has changed; (6) *Object Prediction (OP)*: Predicting what will happen to a certain object.

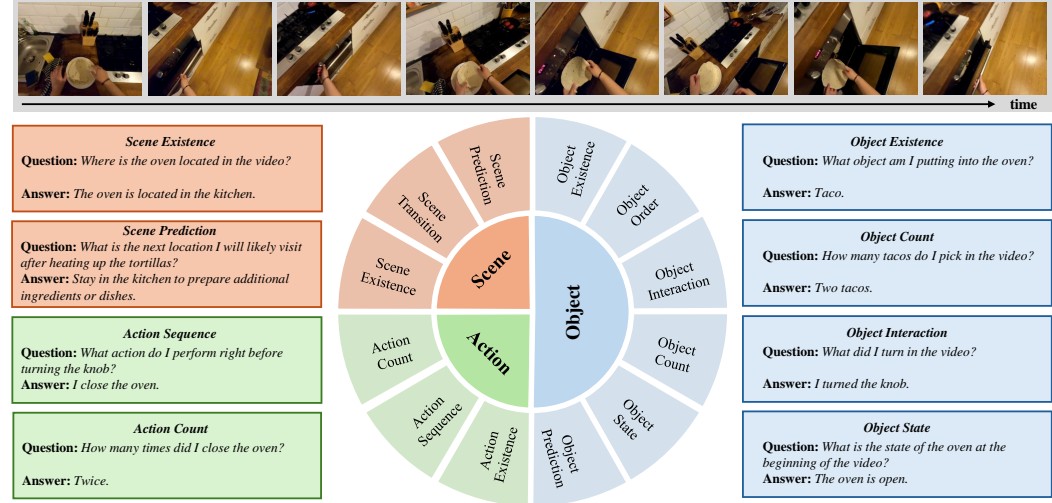

Figure 2: Case of video question answering.

- **Action.** Egocentric videos emphasize events that involve interactions with "myself". Since action prediction is important and has become a standard task in Embodied AI, we will elaborate on it in Sec. 3.2. We divide the action category into three fine-grained dimensions: (1) *Action Existence (AE)*: Determining whether an action occurs; (2) *Action Sequence (AS)*: Identifying the sequence of occurred actions; (3) *Action Count (AC)*: Counting the frequency of occurred actions.

- **Scene.** Perceiving scenes from a first-person perspective is essential for interacting with the environment. The constant movements in egocentric videos makes describing object positions challenging, requiring environmental context integration. Specifically, we design three dimensions: (1) *Scene Existence (SE)*: determining whether the video is in a certain scene; (2) *Scene Transition (ST)*: Identifying transitions between scenes; (3) *Scene Prediction (SP)*: Predicting the next scene.

**Task Format.** Two mainstream methods for video question-answering include *multiple-choice* and *open-ended* question-answering. Open-ended text generation is more natural and practical for real-world applications than multiple-choice, which can be challenging to design distractors without inherent shortcuts. Therefore, we primarily adopt open-ended text generation for our traditional video question-answering tasks.

- **Open-Ended Question-Answering.** Given an egocentric video $i$ along with a question $q_i$, the model is asked to generate responses $r_i$ in free-text form, akin to human communication. The generate answer $r_i$ is then compared to its corresponding ground-truth response $r_i^{gt}$.

**Metrics.** Traditional metrics (Chen et al., 2019; Papineni et al., 2002) fail to accurately assessing semantic similarity. Follwing Zheng et al. (2024b), we use API-based LLMs (*Acc-VQA*) as automatic evaluators. These evaluators have shown high correlation with human labels (Zheng et al., 2024b; Cheng et al., 2024), making them reliable substitutes for human assessment.

- **Acc-VQA.** Given the limitations of traditional metrics, we use API-based LLMs $g(\cdot)$ with superior reasoning abilities to evaluate open-ended answers. Specifically, we assign the score $g(\hat{r}_i, r_i)$ as 0 (wrong), 0.5 (partially correct), or 1 (correct) to the generated response $\hat{r}_i$ with reference to the question $q_i$ and the corresponding ground-truth response $r_i$. The performance of benchmark $\mathcal{D}$ is then computed by averaging all scores as follows:

$$\text{Acc-VQA} = \frac{1}{|\mathcal{D}|}\sum_{i=1}^{|\mathcal{D}|} g(\hat{r}_i, r_i), \ g(\hat{r}_i, r_i) = \begin{cases} 1 & \text{correct} \\ 0.5 & \text{partially correct} \\ 0 & \text{incorrect} \end{cases} \quad (1)$$

## 3.2 HIERARCHY PLANNING

Recently, a hierarchy planning framework (Ahn et al., 2022; Singh et al., 2023; Vemprala et al., 2024) has been proposed to combine foundation models and traditional methods in Embodied AI. Foundation models serve as planners, decomposing high-level goals (e.g., "*cook salmon*") into mid-level steps (e.g., "# put salmon in the microwave') or low-level atomic actions (e.g., "`find`(*microwave*)"). Although EgoPlan-Bench (Chen et al., 2023c) explores planning from a first-person perspective, it only considers decomposing high-level goal into mid-level steps and uses a multiple-choice format, which is less natural.

Figure 3: Case of hierarchy planning.

**Task Format.** As illustrated in Fig. 3, we design two types of planning tasks: high-level goal to mid-level step (*High-to-Mid*), and mid-level step to low-level action (*Mid-to-Low*).

- **High-to-Mid.** Given an egocentric video $i$ with historical and current observations, a high-level goal $G_i$, MLLMs are required to generate the next step $\hat{s}_i$ in free-text format. This generated step is then compared to the ground-truth step $s_i$ that follows the provided video. We adopt a step-by-step format rather than directly generating the entire long-term plan because our focus is on evaluation rather than method development.

- **Mid-to-Low.** Given a pre-defined set of low-level atomic actions $\mathcal{A}$ that encompasses common functions in daily human activities, an egocentric video $V_i$, and the ground-truth of a mid-level step $s_i$ that is yet to be complete, MLLMs are required to generate the trajectory of low-level actions $\hat{\mathcal{T}}_i = (\hat{a}_1, \cdots, \hat{a}_n)$ using functions from $\mathcal{A}$ to complete the mid-level step. The corresponding ground-truth trajectory of actions that appeared after the provided video is $\mathcal{T}_i = (a_1, \cdots, a_m)$.

**Metrics.** Considering the difficulty of hierarchical planning tasks, we directly use API-based LLMs to compute accuracy (*Acc-H2M* and *Acc-M2L*). However, these metrics are a trade-off due to the challenges of evaluation video planning tasks. We will introduce an advanced version in future work, as discussed in Sec. 6.

- **Acc-H2M.** For the High-to-Mid task, we use API-based LLMs $g(\cdot)$ to compute the similarity score $g(\hat{s}_i, s_i)$ between the generated step $\hat{s}_i$ and the ground truth $s_i$ for the benchmark $\mathcal{D}$. We assign the score as 0 (wrong), 0.5 (partially correct), or 1 (correct), similar with Eq. 1.

- **Acc-M2L.** For the Mid-to-Low task, which is akin to tool learning (Guo et al., 2024; Qin et al., 2023) by calling low-level functions and evaluating the success rate, we also use API-based LLMs to determine the completion status. We assign the score $g(\hat{\mathcal{T}}, \mathcal{T})$ to compute the similarity between the generate action trajectory $\hat{\mathcal{T}}$ and the ground-truth trajectory $\mathcal{T}$, using a scale from 0 to 10 to increase the degree of differentiation. The scoring method is otherwise similar to Eq. 1.

## 3.3 VISUAL GROUNDING

Natural language is effective for communication but cannot be directly grounded in the real world. Visual grounding (Peng et al., 2023; Chen et al., 2023a; Munasinghe et al., 2023) addresses this by linking language to images or videos, producing pixel-level bounding boxes, masks, or frames. These outputs identify actionable objects (Munasinghe et al., 2023; Zheng et al., 2024a) and provide spatial or temporal information for downstream tasks (Li et al., 2024c; Chiang et al., 2024).

**Task Format.** RefEgo (Kurita et al., 2023) considers object tracking from the first-person perspective but uses an output format suited for conventional computer vision methods rather than MLLMs. To bridge this gap, we design three tasks tailored for different situations, as shown in Fig. 4: *object grounding*, *frame grounding*, and *temporal grounding*.

- **Object Grounding.** Given an egocentric video $i$ and a natural language query $q_i$ for an object, the model must provide a bounding box $B_i = [x_1, x_2, y_1, y_2]$ containing the query object in the last frame of the video. Performance is evaluated by comparing with the ground truth $B_i^{gt} = [x_1^{gt}, x_2^{gt}, y_1^{gt}, y_2^{gt}]$. Notably, the query $q_i$ is based on the entire video, not just the last frame. Accurately locating target objects that appeared earlier is crucial for downstream tasks like manipulation and navigation.

Figure 4: Cases of visual grounding.

- **Frame Grounding.** Given an egocentric video $i$ and a natural language query $q_i$, the model must identify the keyframe $K_i$ containing the required information. This keyframe is compared with the ground-truth keyframe set $\{K_{ij}^{gt}\}$ around the last appearance of the target, as it generally holds the most useful information for the current situation. In embodied scenes, retrieving objects, people, or events from earlier moments is often necessary.

- **Temporal Grounding.** Given an egocentric video $i$ and a natural language query $q_i$, the model must identify the time segments in the video corresponding to the query, represented as $T_i = [l_i, r_i]$, where $0 \leq l_i \leq r_i \leq |V_i|$ and $|V_i|$ is the total number of frames. The ground truth $T_i^{gt}$ follows the same format. Identifying relevant time segments is crucial for understanding event frequency, object trajectories and so on.

**Metrics.** For object grounding and temporal grounding, we use mean intersection over union (*mIoU*) as the uniform metric, named *mIoU-Object* and *mIoU-Temporal*, respectively. These metrics calculate the similarity between the output and ground truth, as their results can be expressed as regions or ranges. For frame grounding, we use mean square error (*MSE*), since the output is an integer.

- **mIoU-Object.** We denote the bounding box output as $\hat{B}_i$ and the ground truth as $B_i$, where $i$ represents a sample. The similarity in the benchmark $\mathcal{D}$ is calculated using mIoU as follows.

$$\text{mIoU-Obj} = \frac{1}{|\mathcal{D}|} \sum_{i=1}^{|\mathcal{D}|} \frac{|\hat{B}_i \cap B_i|}{|\hat{B}_i \cup B_i|} \tag{2}$$

- **Acc-Frame.** Given the keyframe index $\hat{k}_i$ produced by the model and its corresponding ground truth set $\mathcal{K}_i$, we can calculate the accuracy in the benchmark dataset $\mathcal{D}$ as follows. Here, $\chi(\cdot)$ is an indicator function that equals 1 if $\hat{k}_i$ in $\mathcal{K}_i$, and 0 otherwise.

$$\text{Acc-Frame} = \frac{1}{|\mathcal{D}|} \sum_{i=1}^{|\mathcal{D}|} \chi_{\mathcal{K}_i}\left(\hat{k}_i\right) \tag{3}$$

- **mIoU-Temporal.** We denote the time interval covered by the model output as $T_i$ and the ground truth as $T_i^{gt}$, where $i$ represents a video sample. Similarly, we calculate the similarity in the benchmark $\mathcal{D}$ using the same method as in Eq. 2.

## 3.4 REWARD MODELING

In Embodied AI, designing reward functions for human activities is challenging due to accuracy and diversity requirements. Foundation Models, with their superior commonsense and reasoning capabilities, can serve as reward models. There are three main approaches: (1) Using a sparse proxy reward function with a binary score (Kwon et al., 2023); (2) Computing similarity between action phrases and images (Di Palo et al., 2023; Rocamonde et al., 2023); (3) Generating code to translate task semantics into reward functions (Yu et al., 2023; Ma et al., 2023). This paper focuses on the first approach for video data.

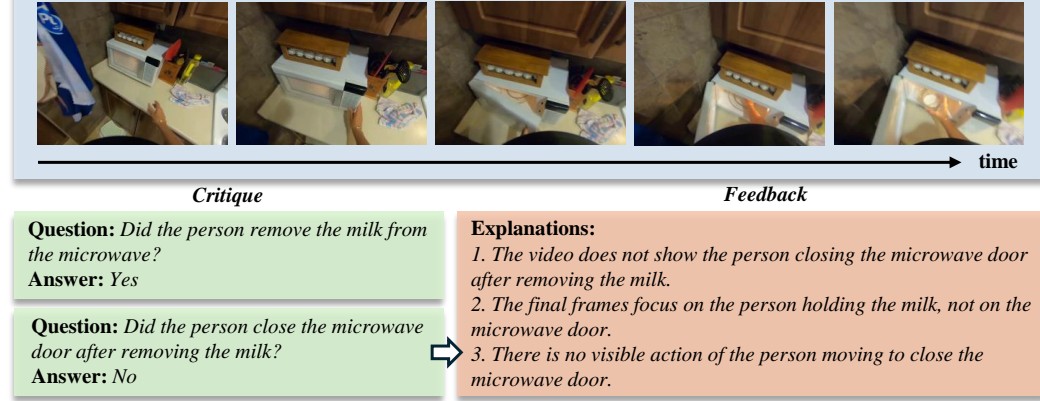

**Critique**

**Question:** *Did the person remove the milk from the microwave?*
**Answer:** *Yes*

**Question:** *Did the person close the microwave door after removing the milk?*
**Answer:** *No*

**Feedback**

**Explanations:**
*1. The video does not show the person closing the microwave door after removing the milk.*
*2. The final frames focus on the person holding the milk, not on the microwave door.*
*3. There is no visible action of the person moving to close the microwave door.*

Figure 5: Case of reward modeling.

**Task Format.** As a reward model, MLLMs should observe the video to determine the completion status of the target motion. If the action is not completed, the model should further provide fine-grained feedback to help achieve the goal (Wang et al., 2020; Cheng et al., 2023). We design two tasks, as shown in Fig. 5: *critique*, and *feedback*.

- **Critique**. Given an action-specific egocentric video $i$ and its corresponding natural language action description $a_i$, the reward model needs to directly generate a binary answer $\hat{y}_i$ (i.e., Yes or No) that indicates whether the action has been completed in the observed video.

- **Feedback**. Given an uncompleted action $a_i$ in the $i$-th egocentric video, the reward model provides fine-grained feedback $\hat{r}_i$ on why the action is not completed based on current observations, compared to the ground-truth references $\mathcal{R}_i = \{r_i^1, r_i^2, r_i^3\}$. This feedback guides and corrects downstream models to learn the policy for completing the target action.

**Metrics.** We use the following accuracy metrics to assess performance in critique and feedback tasks (*Acc-Critique* and *Acc-Feedback*) for the reward modeling tasks.

- **Acc-Critique**. We compare the generated critique $\hat{y}_i$ with its ground-truth label $y_i$ in the benchmark $\mathcal{D}$. The boolean function $\mathbb{I}(\cdot)$ returns one for each identical labels and zero otherwise.

$$\text{Acc-Cri} = \frac{1}{|\mathcal{D}|} \sum_{i=1}^{|\mathcal{D}|} \mathbb{I}(\hat{y}_i = y_i) \tag{4}$$

- **Acc-Feedback**. To assess the similarity between the generated feedback $\hat{r}_i$ and the set of reference feedback $\mathcal{R}_i$, we use evaluator LLM $g(\cdot)$ to assign a score of 0 (wrong), 0.5 (partially correct) or 1 (correct), similar to Eq. 1.

## 4 DATA COLLECTION IN *VidEgoThink*

Recent releases of egocentric video datasets (Grauman et al., 2022; 2024; Huang et al., 2024) have advanced Embodied AI. We use the popular Ego4D dataset (Grauman et al., 2022) for our benchmark. Ego4D-v2[1] contains 3,900 hours of 9,611 egocentric videos with diverse human annotations. To avoid data leakage, we select videos from the validation dataset. However, due to the video length limitations of MLLMs, the lengthy Ego4D videos, ranging from tens of minutes to over an hour, are unsuitable. Additionally, manually labeling question-answering data requires significant human effort. To address these problems, we design strategies to automatically clip the videos to appropriate lengths and generate corresponding question-answer pairs. To prevent the VidEgoThink benchmark from being compromised through prompt engineering, the detailed prompts used for automatic annotation construction will not be released. The statistics of each task in VidegoThink are presented in Table 4 in Appendix B.

**Video Question-Answering.** To construct this benchmark, we integrate *Narration* data, capturing interactions between the camera wearer and the environment, focusing on *object*, *action*, and *scene*. We develop specific prompts, combined with the narrations, as inputs for GPT-4o tailored to each fine-grained dimension. GPT-4o then generate diverse question-answering pairs for these dimensions. Due to the noise in generated instances and the cost of API-based evaluation, three human

---

[1]https://ego4d-data.org/docs/updates/

annotators filter them to ensure quality and diversity, selecting the most representative examples. Finally, we totally collect 600 instances with 50 examples per fine-grained dimensions.

**Hierarchy Planning.** We use existing human annotations in Ego4D with goals-steps-substeps labels to construct our hierarchical data. For video inputs, we use from 00:00 to the start time of the current step for both high-to-mid and mid-to-low subtasks. In the high-to-mid task, high-level goals serve as inputs and corresponding mid-level step as labels. Steps requiring numerous low-level actions and exceeding 180 seconds are decomposed into essential substeps. Next, we use the ground-truth mid-level step and its *Narration* as potential low-level atomic actions. To align with Embodied AI controller, GPT-4o converts narrations (e.g., "*C cuts a mango with a knife*") into function calls (e.g., "cut(*mango*, *knife*)") and merges semantically similar functions. To ensure MLLMs understand the available low-level functions and their usage, we apply GPT-4o to generate their documentation. After filtering by three annotators, we obtain 598 clipped videos and instances for both tasks, with the mid-to-low task comprising 74 atomic actions.

**Visual Grounding.** *Visual Queries* in Ego4D includes queries about objects and their tracks in the video, represented as frames with bounding boxes. We use these annotations to collect object grounding and frame grounding subtasks. For object grounding, given a clipped video and its annotations, we select the video from the beginning to the last annotated frame. We construct a prompt with the *Narration* in this segment for GPT-4o to generate a query. The answer is the bounding box annotation of the object in the final frame. In frame grounding, the video input spans from the start of the clipped video to either the "query_video_frame" annotated in *Visual Queries* or the end frame of the clip. We prompt GPT-4o using the object name and narrations within the time segment to generate a specific description of the frames containing the object. All annotated frames in the input video are considered the answer. Considering that step-substep annotations in *Goal-Step* include temporal information, we primarily use these clipped videos. By providing annotations and prompts to GPT-4o, we obtain a specific description of the selected sub-step as the query and the temporal interval of the sub-step as the answer. Finally, we obtain 369 instances for object and frame grounding, and 735 instances for temporal grounding.

**Reward Modeling.** Our clipped videos in the *hierarchy planning* task contain entire mid-level steps, which we use to construct the reward modeling dataset. We label the original complete videos as positive instances. For negative instances, we employ two strategies: (1) using GPT-4o to generate questions where the action is similar but different from the video content; (2) manually crop each video clip to 60%–80% of its original length to ensure the action remains unfinished. Each negative sample includes three feedback demonstrating the incomplete action. Considering narrations often lack detailed descriptions to determine whether an action is complete, we employ FFmpeg[1] to extract keyframes from each clipped video. Then, we use GPT-4o to generate feedback from different aspects for negative instances based on step annotations and the extracted keyframes. After filtering by three annotators, we obtain 963 and 638 instances for critique and feedback tasks.

## 5 EXPERIMENTS

In this section, we mainly introduce our extensive adopted models, including API-based models, a series of open-source image-based and video-based MLLMs. The detailed information of all these MLLMs are presented in Appendix C and the prompts for both inference and evaluation are shown in Appendix D. Furthermore, we summarize the experimental results for different tasks, and their correpsonding case studies are illustrated in Appendix E.

### 5.1 MODELS

**API-based Models.** We conduct experiments with the representative GPT-4o (2024-05-13). Since GPT-4o does not support video input, we address this limitation and enhance methodological diversity with the following assessment scheme: (1) *w/ 32 frames*: Select 32 keyframes based on the video context; (2) *w/ 8 frames*: Select 8 keyframes with the same input format as most open-source MLLMs; (3) *w/ captions*: Replace 32 keyframes with its corresponding captions generated by GPT-4o; (4) w/ only-qa: Input only the question without any frames or captions.

---

[1] https://www.ffmpeg.org/

Table 1: Experimental results of video question answering. OE, OO, OI, OC, OS, OP denote object existence, object order, object interaction, object count, object state, object prediction. AE, AS, AC indicates action existence, action sequence, action count. SE, ST, SP denote scene existence, scene transition, scene prediction. The **bold** font denotes the best performance and the underline font denotes the second-best performance.

| Models | Object | | | | | | Action | | | Scene | | | Average |
|---|---|---|---|---|---|---|---|---|---|---|---|---|---|
| | OE | OO | OI | OC | OS | OP | AE | AS | AC | SE | ST | SP | |
| **GPT-4o** w/ only-qa | 13.00 | 0.00 | 12.00 | 6.00 | 31.00 | 23.00 | 25.00 | 4.00 | 2.00 | 18.00 | 6.00 | 20.00 | 13.33 |
| **GPT-4o** w/ captions | 51.00 | 16.00 | 14.00 | 30.00 | 25.00 | 44.00 | 34.00 | 5.00 | 22.00 | 42.00 | **28.00** | 16.00 | 27.25 |
| **GPT-4o** w/ 8 frames | 51.00 | 16.00 | **30.00** | 33.00 | **35.00** | **45.00** | 38.00 | **25.00** | 22.00 | 43.00 | 23.00 | **24.00** | **32.83** |
| **GPT-4o** w/ 32 frames | **52.00** | 18.00 | **30.00** | **35.00** | 32.00 | 40.00 | **39.00** | 20.00 | 24.00 | **46.00** | 20.00 | 18.00 | 31.17 |
| **mPLUG-Owl2-llama2-7B** | 29.00 | 6.00 | 15.00 | 30.00 | 10.00 | 16.00 | 28.00 | 8.00 | 28.00 | 20.00 | 10.00 | 6.00 | 17.17 |
| **Qwen-VL-7B-Chat** | 41.00 | 7.00 | 13.00 | 33.00 | 14.00 | 30.00 | 17.00 | 3.00 | 27.00 | 16.00 | 13.00 | 10.00 | 18.67 |
| **LLaVA-1.5-7B** | 46.00 | 7.00 | 17.00 | 34.00 | 22.00 | 24.00 | 25.00 | 1.00 | 14.00 | 20.00 | 13.00 | 16.00 | 19.92 |
| **LLaMA-Adapter-V2-7B** | 48.00 | 5.00 | 26.00 | 17.00 | 19.00 | 39.00 | 14.00 | 9.00 | 35.00 | 24.00 | 10.00 | 16.00 | 21.80 |
| **LWM-Chat-32k-Jax-7B** | 42.00 | 3.00 | 20.00 | 12.00 | 10.00 | 11.00 | 20.00 | 4.00 | 21.00 | 27.00 | 9.00 | 5.00 | 15.33 |
| **TimeChat-7B** | 42.00 | 5.00 | 15.00 | 21.00 | 11.00 | 23.00 | 20.00 | 4.00 | 20.00 | 31.00 | 14.00 | 14.00 | 18.33 |
| **GroundingGPT-7B** | 43.00 | 3.00 | 20.00 | 30.00 | 10.00 | 23.00 | 22.00 | 4.00 | 32.00 | 23.00 | 19.00 | 14.00 | 20.25 |
| **InternVL2-8B** | 43.00 | 16.00 | 21.00 | 18.00 | 20.00 | 27.00 | 19.00 | 4.00 | 15.00 | 37.00 | 17.00 | 12.00 | 20.75 |
| **InternLM-XComposer2.5-7B** | 36.00 | 6.00 | 24.00 | 22.00 | 19.00 | 34.00 | 30.00 | 2.00 | 30.00 | 31.00 | 11.00 | 12.00 | 21.42 |
| **Video-LLaVA-7B** | 44.00 | 8.00 | 19.00 | 34.00 | 15.00 | 30.00 | 18.00 | 3.00 | **38.00** | 28.00 | 11.00 | 11.00 | 21.58 |
| **PG-Video-LLaVA-7B** | 49.00 | 5.00 | 21.00 | 15.00 | 23.00 | 37.00 | 25.00 | 3.00 | 16.00 | 35.00 | 18.00 | 20.00 | 22.25 |
| **mPLUG-Owl3-7B** | 32.00 | 7.00 | 26.00 | 13.00 | 33.00 | 34.00 | 18.00 | 6.00 | 36.00 | 37.00 | 23.00 | 10.00 | 22.92 |
| **MiniCPM-V-2.6-8B** | 48.00 | 12.00 | 28.00 | 16.00 | 25.00 | 42.00 | 31.00 | 11.00 | 15.00 | 42.00 | 23.00 | 18.00 | 25.92 |
| **Qwen2-VL-7B-Instruct** | 36.00 | **19.00** | 28.00 | 28.00 | 28.00 | 43.00 | 24.00 | 9.00 | 20.00 | **48.00** | 24.00 | 20.00 | 27.25 |

**Open-Source MLLMs.** We consider both image-based and video-based MLLMs. For image-based MLLMs, we select those that demonstrated strong performance in EgoThink (Cheng et al., 2024). Additionally, we comprehensively choose the most popular and high-performance video-based MLLMs, including a series of general models and three grounding-specific models.

## 5.2 RESULTS

**Video Question-Answering.** The results of the video question-answering task are shown in Table 1 and Table 2. MLLMs perform poorly, with a best average accuracy of 32.82% across all dimensions (35.00% for object, 28.33% for action, and 26.33% for scene elements), indicating struggles with egocentric video question-answering. GPT-4o with 8 frames performs better than with 32 frames but still underperforms compared to some open-source video MLLMs in certain dimensions. Two probable reasons are: (1) GPT-4o's sensitivity to privacy policies for indoor videos, causing it to refuse more questions given more images; (2) insufficient information from extracted keyframes. GPT-4o with captions sometimes matches or surpasses the 8 or 32-frame setups in scene transitions, but performs poorly in object interaction and action sequence dimensions, indicating that captions provide sufficient high-level abstraction but lack detailed low-level action information. We regard the GPT-4o with only-qa as a baseline to demonstrate state-of-the-art performance using only question-answering pairs without any vision information. All other MLLMs perform better than the average accuracy of GPT-4o with only-qa, showing that our benchmark indeed requires vision information to solve these problems. Open-source video-based MLLMs generally surpass image-based MLLMs, highlighting the need for full video information, especially in dynamic dimensions. Among these, Qwen2-VL-7B-Instruct achieves the best performance, even surpassing GPT-4o in two dimensions and achieving the second-best performance in three dimensions.

**Hierarchy Planning.** The hierarchy planning results are shown in Table 2, with the average video duration being 1008.26 seconds. In the High-to-Mid task, GPT-4o series models and image-based MLLMs, which process multiple or single images, lack sufficient information to determine the entire progress and predict the next step. Hence, increasing the total number of frames significantly improves performance. For video-based models, the best performance of MiniCPM is comparable to the state-of-the-art performance of GPT-4o with 32 frames but still performs poorly, indicating significant room for improvement. For the Mid-to-Low task, the most notable phenomenon is that GPT-4o series models significantly outperform open-source MLLMs, which achieve about 0.05 accuracy. The main reason behind this phenomenon is the limited long-context capability and instruction-following capability of open-source MLLMs. We can only provide them with a compressed function document, and they often do not generate answers following the output format.

**Visual Grounding.** Visual grounding tasks involve identifying specific objects, frames, or temporal segments within a video. API-based and image-based MLLMs abandon this information after extracting keyframes, necessitating the use of open-source video-based MLLMs for performance

Table 2: Experimental results of video question answerng, hierarchy planning, visual grounding, and reward modeling tasks. The **bold** font denotes the best performance and the underline font denotes the second-best performance.

| Models | Video Question Answering | | | Hierarchy Planning | | Visual Grounding | | | Reward Modeling | |
|---|---|---|---|---|---|---|---|---|---|---|
| | Object | Action | Scene | High-to-Mid | Mid-to-Low | Object | Frame | Temporal | Critique | Feedback |
| **GPT-4o** w/ only-qa | 14.17 | 10.33 | 14.67 | 8.86 | 32.56 | - | - | - | 48.46 | 6.81 |
| **GPT-4o** w/ captions | 30.00 | 20.33 | 28.67 | 9.53 | 33.65 | - | - | - | 58.82 | 14.58 |
| **GPT-4o** w/ 8 frames | **35.00** | **28.33** | **30.00** | 12.04 | **35.47** | - | - | - | 58.74 | 33.46 |
| **GPT-4o** w/ 32 frames | 34.50 | 27.67 | 26.33 | **14.97** | 35.08 | - | - | - | **59.39** | **34.64** |
| **mPLUG-Owl2-llama2-7B** | 17.67 | 21.33 | 12.00 | 5.77 | 0.00 | - | - | - | 41.26 | 1.56 |
| **Qwen-VL-7B-Chat** | 23.00 | 15.67 | 13.00 | 10.79 | 0.04 | - | - | - | 49.19 | 4.08 |
| **LLaVA-1.5-7B** | 25.00 | 13.33 | 16.33 | 2.59 | 0.01 | - | - | - | 53.72 | 3.53 |
| **LLaMA-Adapter-V2-7B** | 25.67 | 19.33 | 16.67 | 4.59 | 0.03 | - | - | - | 39.64 | 2.89 |
| **LWM-Chat-32k-Jax-7B** | 16.33 | 15.00 | 13.67 | 1.33 | 0.00 | 0.00 | 0.00 | 0.00 | 22.09 | 0.00 |
| **TimeChat-7B** | 19.50 | 14.67 | 19.67 | 3.85 | 0.01 | 0.00 | 0.00 | 14.56 | 47.25 | 0.57 |
| **GroundingGPT-7B** | 21.50 | 19.33 | 18.66 | 5.69 | 0.05 | **0.76** | 0.54 | 0.44 | 51.13 | 2.19 |
| **InternVL2-8B** | 24.17 | 12.67 | 22.00 | 3.34 | 0.05 | 0.09 | 0.00 | 6.87 | 52.67 | 0.71 |
| **InternLM-XComposer2.5-7B** | 23.50 | 20.67 | 18.00 | 9.62 | 0.04 | 0.00 | 0.54 | 3.50 | 51.41 | 8.23 |
| **PG-Video-LLaVA-7B** | 25.00 | 14.67 | 24.33 | 5.35 | 0.00 | 0.08 | 0.00 | **16.18** | 48.30 | 6.27 |
| **mPLUG-Owl3-7B** | 24.17 | 20.00 | 23.33 | 12.29 | 0.03 | 0.00 | 0.00 | 0.00 | 50.00 | 9.09 |
| **MiniCPM-V-2.6-8B** | 28.50 | 19.00 | 27.67 | 14.13 | 0.06 | 0.35 | **1.63** | 11.30 | 51.54 | 13.09 |
| **Qwen2-VL-7B-Instruct** | 30.33 | 16.00 | 27.67 | 9.88 | 0.00 | 0.00 | 0.00 | 0.00 | 49.03 | 4.62 |

assessment. Due to the new design of object and frame grounding tasks, these MLLMs are not yet optimized for these formats, leading to generally poor performance. It is understandable that object grounding in a single image remains a challenging task for MLLMs, even more so within a video context. For temporal grounding, some MLLMs especially trained for this task achieve relatively high scores, with PG-Video-LLaVA scoring 16.18. Surprisingly, MiniCPM performs well across all grounding dimensions, despite not being specially trained for these tasks. Although the performances of MLLMs are poor, we believe these tasks have a significant impact on downstream tasks in Embodied AI and deserve more attention.

**Reward Modeling.** As shown in Table 2, the critique task is a binary classification task with a random guess baseline of 50%. Therefore, the overall performance of MLLMs is suboptimal, with the best accuracy reaching only 59.39%, indicating that MLLMs struggle to determine whether a task has been completed. For the feedback task, GPT-4o with 8 frames (33.46%) and 32 frames (34.64%) significantly outperforms the best results from other API-based methods (14.58%) and open-source MLLMs (13.09%). This demonstrates that generating feedback requires more fine-grained visual information not present in captions and superior reasoning capability.

## 6 CONCLUSION

In this paper, we introduce VidEgoThink, a comprehensive benchmark designed to evaluate egocentric video understanding across four critical functions in Embodied AI. Our assessment of popular API-based and open-source MLLMs reveals that these models still face significant challenges in processing egocentric videos. Although GPT series models perform relatively better, they exhibit notable deficiencies in certain areas, highlighting the need for further improvements and optimizations. VidEgoThink underscores the limitations of current MLLMs in handling first-person perspective data, thereby indicating directions for future research and advancements

**Limitations.** VidEgoThink is the first to propose four tasks for assessing egocentric video understanding in MLLMs for Embodied AI. However, it has limited data diversity and immature evaluation methods, particularly in hierarchy planning and reward modeling. Future work should improve these aspects and address the high costs of human annotation and API-based evaluations, which limit the number of question-answer pairs. We plan to expand the benchmark and develop egocentric foundation models for robotics.

**Broader Impacts.** Two key areas for the future of Embodied AI are *egocentric video* and *multimodal large language models* On the one hand, our real world cannot be mapped to virtual simulators exactly the same way. Real-world environments cannot be exactly replicated in virtual simulators, making egocentric video a preferred method for collecting action data, especially with the rise of smart glasses and humanoid robots. Learning from egocentric video is crucial for future advancements. Although end-to-end MLLMs for Embodied AI are still an open research question, we believe a hierarchical system that uses vision-language models for perception and cognition is an emerging paradigm. Ideal foundation models should function in the real world, capable of thinking, understanding, and interacting like humans.

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

# A COMPARISON TO RECENT BENCHMARKS

Table 3: Comparison of recent evaluation benchmarks of multimodal large language models and our proposed benchmark VidEgoThink. VQA/HP/VG/RM indicate visual question answering, hierarchy planning, visual grounding, and reward modeling. Existing/Handcraft/Automatic denote the way of collecting data, including existing dataset, manual annotation, and automatic generation.

| Benchmark | Comprehensive Capabilities | View | | Task Type | | | | Data Source | Average Length | Total Size |
|---|---|---|---|---|---|---|---|---|---|---|
| | | Observe | Interact | VQA | HP | VG | RM | | | |
| ActivityNet-QA | ✗ | ✓ | ✗ | ✓ | ✗ | ✗ | ✗ | Handcraft | 180s | 58,000 |
| SEED-Bench-2 | ✓ | ✓ | ✗ | ✓ | ✗ | ✗ | ✗ | Handcraft | - | 24,000 |
| AutoEval-Video | ✓ | ✓ | ✗ | ✓ | ✗ | ✗ | ✗ | Handcraft | 14.58s | 327 |
| Video-Bench | ✓ | ✓ | ✗ | ✓ | ✗ | ✗ | ✗ | Existing | - | 15,000 |
| Perception Test | ✗ | ✓ | ✗ | ✓ | ✗ | ✓ | ✗ | Handcraft | 23s | 11,600 |
| OpenEQA | ✗ | ✓ | ✗ | ✓ | ✗ | ✗ | ✗ | Handcraft | - | 1,600 |
| MVBench | ✓ | ✓ | ✓ | ✓ | ✗ | ✗ | ✗ | Existing | (5s, 35s) | 4,000 |
| EgoVQA | ✗ | ✓ | ✓ | ✓ | ✗ | ✗ | ✗ | Handcraft | (20s, 100s) | 520 |
| EgoThink | ✓ | ✗ | ✓ | ✓ | ✓ | ✗ | ✗ | Handcraft | - | 700 |
| EgoTaskQA | ✗ | ✗ | ✓ | ✓ | ✗ | ✗ | ✗ | Automatic | 25s | 40,000 |
| EgoPlan-Bench | ✗ | ✗ | ✓ | ✗ | ✓ | ✗ | ✗ | Automatic | - | 3,400 |
| EgoSchema | ✗ | ✗ | ✓ | ✓ | ✗ | ✗ | ✗ | Automatic | 180s | 5,000 |
| **VidEgoThink** (Ours) | ✓ | ✓ | ✓ | ✓ | ✓ | ✓ | ✓ | Automatic | 270.74s | 4,993 |

# B STATISTICS OF VIDEGOTHINK

In this section, we present detailed statistics of the VidEgoThink benchmark, including information about videos and question-answering pairs.

- **Number of original videos (#Original):** The total number of original, entire videos in the Ego4D dataset.
- **Number of clipped videos (#Clipped):** The total number of clipped videos of moderate duration from the original videos.
- **Duration:** The average duration (in seconds) of the clipped videos.
- **Number of instance (#Instance):** The total number of video question-answer pairs in each task.
- **Question length (LenQ):** The average length of the questions, measured in words.
- **Answer legnt (LenA):** The average length of the answers, measured in words.
- **Question Type (TypeQ):** The total count of various types of questions.
- **Number of Scenes (#Scene):** The total number of types of scenes officially annotated by Ego4D.

Table 4: The statistics of videos across different benchmarks. Duration denotes the average time duration in second of all videos. LenQ and LenA indicate that the average length of questions and answers in the word level. TypeQ denotes the type of questions.

| Benchmark | Subtask | Video | | | Question-Answering | | | | #Scene |
|---|---|---|---|---|---|---|---|---|---|
| | | #Original | #Clipped | Duration | #Instance | LenQ | LenA | TypeQ | |
| **Video Question Answering** | Object | 29 | 57 | 23.71 | 300 | 10.88 | 7.13 | 5 | 9 |
| | Action | 39 | 78 | 24.56 | 150 | 10.85 | 4.72 | 4 | 9 |
| | Scene | 45 | 82 | 21.91 | 150 | 11.46 | 8.34 | 4 | 9 |
| **Hierarchy Planning** | High-to-Mid | 76 | 598 | 1008.26 | 598 | 16.5 | 5.18 | 1 | 9 |
| | Mid-to-Low | 76 | 598 | 1008.26 | 598 | 22.12 | 6.02 | 1 | 9 |
| **Visual Grounding** | Object | 41 | 88 | 119.05 | 220 | 22.60 | - | 1 | 25 |
| | Frame | 65 | 147 | 139.57 | 368 | 23.01 | - | 1 | 25 |
| | Temporal | 69 | 416 | 68.90 | 735 | 82.40 | - | 1 | 8 |
| **Reward Modeling** | Critique | 76 | 963 | 16.60 | 1236 | 11.21 | 1.00 | 1 | 9 |
| | Feedback | 74 | 638 | 15.08 | 638 | 19.24 | 53.06 | 1 | 9 |

## C    MODELS HUB

In this section, we briefly introduce the open-source MLLMs used for evaluation. The important components of all these open-source MLLMs are shown in Table 5.

### C.1    OPEN-SOURCE IMAGE-BASED MLLMS

The brief introduction of all open-source imaged-bsaed MLLMs is listed below:

- **LLaVA-1.5** (Liu et al., 2024b) utilizes academic task data and replaces the linear visual language connector with a two-layer MLP connector.
- **LLaMA-Adapter V2** (Gao et al., 2023b) proposes an early fusion strategy that effectively adapts LLaMA (Touvron et al., 2023) to visual instruction models.
- **Qwen-VL-Chat** (Bai et al., 2023a) employs a single-layer cross-attention with random initialization, trained with approximately 1.5 billion image-text pairs, and aligns with human interaction.
- **mPLUG-Owl2** (Ye et al., 2024b) integrates shared functional modules to promote modality collaboration and includes a modality-adaptive module to preserve modality-specific features.

### C.2    OPEN-SOURCE VIDEO-BASED MLLMS

The brief introduction of all open-source video-bsaed MLLMs is listed below:

- **InternVL2** (Chen et al., 2023d) builds on InternVL's QLLaMA progressive alignment strategy. It optimizes vision-language alignment while scaling up the language model in stages, starting small and expanding gradually, with data refined from coarse to fine.
- **MiniCPM-V-2.6** (Yao et al., 2024) utilizes the adaptive visual encoding mechanism of LLaVA-UHD (Xu et al., 2024) and various end-side optimizations to compress the multimodal model.
- **Qwen2-VL** (Bai et al., 2023b; team, 2024) has been upgraded with Naive Dynamic Resolution and Multimodal Rotary Position Embedding (M-ROPE) technologies, improving its multimodal data processing and understanding capabilities.
- **InternLM-XComposer-2.5** (Zhang et al., 2024) introduces RoPE extrapolation for long-context handling, ultra-high resolution understanding, fine-grained video comprehension, and multi-turn multi-image dialogue, and extra LoRA parameters for advanced text-image composition.
- **Video-LLaVA** (Lin et al., 2023) proposes a unified visual representation method that aligns images and videos within the language feature space. This approach enhances multimodal interactions and leverages a mixed dataset of images and videos to mutually improve each modality.
- **LWM** (Liu et al., 2024a) uses Blockwise RingAttention and masked sequence packing to manage long video and language sequences, enabling training on contexts up to 1 million tokens for better multimodal understanding.
- **mPLUG-Owl3** (Ye et al., 2024a) introduces hyper attention blocks to efficiently integrate vision and language into a shared semantic space, improving long image sequence processing. video benchmarks.
- **PG-Video-LLaVA** (Munasinghe et al., 2023) is a video-based MLLM with pixel-level grounding capabilities. It can also integrate audio to enhance video understanding. Additionally, its modular design enhances flexibility.
- **GroundingGPT** (Li et al., 2024b) effectively enhances the understanding and grounding of fine-grained image, video, and audio modalities through a three-stage, coarse-to-fine training strategy.
- **TimeChat** (Ren et al., 2024) is a time-sensitive multimodal large language model that aligns visual information with specific time frames. It utilizes a sliding video Q-Former to adapt to videos of varying lengths.

## D    PROMPT HUBS

To address concerns about potential data breaches through prompts, here we only release the detailed prompts for each task to facilitate inference and evaluation.

Table 5: LM, VM, TM, AM refer to the language module, visual module, temporal module, and alignment module, respectively. CLIP-ViT-L is CLIP module pre-trained on LLaVA, while CLIP-ViT-G is the CLIP module pre-trained on LAION.

| Model | LM | VM | TM | AM | Model Size | Training Data Image/Video-Text | Instruction |
|---|---|---|---|---|---|---|---|
| **API-based Model** | | | | | | | |
| GPT-4o | | | Unknown | | | | |
| **Image-based MLLMs** | | | | | | | |
| mPLUG-Owl2 | LLaMA | CLIP-ViT-L | - | Visual Abstractor | 7B | 1.23M | - |
| Qwen-VL | Qwen | CLIP-ViT-G | - | VL Adapter | 7B | 1.4B | 350K |
| LLaVA 1.5 | LLaMA/Vicuna | CLIP-ViT-L-3 | - | Linear | 7B | 558K | 665K |
| LLaMA-Adapter v2 | LLaMA | CLIP-ViT-L | - | Linear | 7B | 567K | 52K |
| **Video-based MLLMs** | | | | | | | |
| LWM | LLaMA2 | VQGAN | - | - | 7B | 1.01B | 519K |
| TimeChat | LLaMA2 | CLIP-ViT-G | Time-aware Frame Encoder | Sliding Video Q-Former | 7B | - | 177K |
| GroundingGPT | Vicuna-v1.5 | CLIP-ViT-L | position encoding | MLP | 7B | >1.3M | >770K |
| InternVL2 | InternLM2.5 | InternViT-300M-448px | - | QLLaMA | 8B | 10B | - |
| InternLM-XComposer2.5 | InternLM2 | CLIP-ViT-L | - | Partial-LoRA | 7B | - | - |
| PG-Video-LLaVA | Vicuna-v1.5 | CLIP-ViT-L-3 | Grounding Module | MLP | 7B | - | 100K |
| mPLUG-Owl3 | Qwen2 | SigLip-400M | MI-RoPE | Linear | 8B | >1.7M | >1M |
| MiniCPM-V2.6 | Qwen2 | SigLip-400M | - | Adaptive Visual Encoding | 8B | 570M | 3M |
| Qwen2-VL | Qwen2 | ViT | M-RoPE | 3D-conv | 8B | $1.4T_{tokens}$ | - |

## D.1 MODEL INFERENCE PROMPTS

As an example, we list the general prompts for 8 frames, 32 frames and open-source MLLMs. The inference type of "caption" for GPT series models will add a prompt "*Here is the captions of the video: {caption}.*" after the sentence "*Imagine you are the camera wearer (I) who recorded the video*". For the inference type of "only-qa", we delete the prompt "*Imagine you are the camera wearer (I) who recorded the video*".

- **Video Question Answering:** *Imagine you are the camera wearer (I) who recorded the video. Please directly answer the question as short as possible. Question: {question} Short answer:*

- **High-to-Mid in Hierarchy Planning:** *Imagine you are the camera wearer (I) who recorded the video. Given the high-level goal (e.g., 'making dumpling') and the current progress video, you need to predict the next mid-level step (e.g., fold dumplings on a cutting board) to achieve the goal. Please directly generate the next one step as short as possible. Question: {question} Short answer:*

- **Mid-to-Low in Hierarchy Planning:** *Imagine you are the camera wearer (I) who recorded the video. Here are a set of actionable functions below.*
[begin of actionable function and documentation]
*{'put': 'put(<arg1>, <arg2>) is used to place an object at a specified or default location. <arg1> refers to the item to be placed, whereas <arg2> is optional and specifies the location where the item should be placed. If <arg2> is omitted, the item is placed in a generic, predefined area.',*
*'grab': 'grab(<arg1>, <arg2>) is used to simulate the action of grasping or picking up objects, especially in a kitchen setting. <arg1> refers to the primary object to be grabbed, while <arg2> is optional and denotes an associated tool or container that aids in handling or processing the primary object.',*
*'talk': 'talk(<arg1>, <arg2>) is used to simulate a conversation scenario with specific entities. <arg1> is mandatory and specifies the primary entity involved in the conversation, such as a 'woman', 'man', or 'person'. <arg2> is optional and typically represents a secondary entity or context within the conversation, providing additional detail or focus.',*
*'close': 'close(<arg1>, <arg2>) is used to encapsulate or seal an item, either partially or completely. <arg1> refers to the object to be closed or covered, and <arg2> is optional, describing the material or object used for closing or covering <arg1>. If <arg2> is omitted, the closing is done without any specified covering.',*
*'adjust': 'adjust(<arg1>, <arg2>) is used to modify the position or settings of objects or items. <arg1> is mandatory and specifies the primary object to adjust, while <arg2> is optional and used for adjustments involving a specific secondary object or location relative to the first.',*
*'arrange': 'arrange(<arg1>, <arg2>) is used to organize objects systematically within a predefined space. <arg1> refers to the items to be arranged, while <arg2> is optional and specifies the area or container where these items will be organized. If <arg2> is omitted, the items are arranged in a default designated space.',*

'open': 'open(<arg1>, <arg2>) is used to manipulate the state of various containers or coverings by opening them. <arg1>refers to the primary object or container that needs to be opened, like a 'pot' or 'drawer'. <arg2>is optional and specifies a secondary descriptor or specific part of the primary object, like 'top' or 'front', indicating a particular method or area of opening.',

'walk': 'walk(<arg1>, <arg2>) is used to move an entity towards a specified location within an environment. <arg1>refers to the primary location or object the entity should head towards, and <arg2>refers to optional additional parameters that provide extra directional or contextual details to refine the movement.',

'empty': 'empty(<arg1>, <arg2>) is used to transfer a specified item from one holding medium to another specified container. <arg1>refers to the item being transferred, while <arg2>is the destination container where the item is moved to.',

'move': 'move(<arg1>, <arg2>) is used to transfer items from one place to another. <arg1>refers to the item that is being moved. <arg2>is optional and specifies where the item should be placed; if omitted, it indicates the item is moved without a specific destination in mind, likely for clearing space or as an intermediate step.',

'push': 'push(<arg1>, <arg2>) is used to initiate a push action on various objects or elements. <arg1>refers to the main object or element to be pushed, and <arg2>is optional and used to specify a particular part or aspect of <arg1>for a more precise push action.',

'clean': 'clean(<arg1>, <arg2>) is used to cleanse various items, which may include food or non-food objects. <arg1>refers to the primary item that requires cleaning, while <arg2>is optional and specifies additional items or the context like the cleaning environment or method. If <arg2>is omitted, the function adapts its operation to effectively clean <arg1>alone.',

'rotate': 'rotate(<arg1>, <arg2>) is used to turn or move an item, typically in a culinary context. <arg1>refers to the item that needs to be rotated. <arg2>is optional and describes the utensil or tool used to facilitate the rotation. If <arg2>is omitted, the item is rotated manually or with a default method.',

'serve': "serve(<arg1>, <arg2>) is used to manage the distribution or placement of items. <arg1>refers to the item to be served or used, and <arg2>is optional, indicating the person or the hand (right or left) that will handle the item. If <arg2>is omitted, the item is handled by default means.',

'shell': 'shell(<arg1>, <arg2>) is used to remove the outer covering from items, typically food-related like seeds, vegetables, and fruits. <arg1>is mandatory and specifies the item from which the shell or outer layer needs removal. <arg2>is optional and indicates any tool that might assist in the shelling process, such as a knife or fork. If <arg2>is omitted, the item is shelled using standard methods.',

'turn_on': 'turn_on(<arg1>, <arg2>, etc) is used to activate one or multiple household or industrial appliances. <arg1>is mandatory and refers to the primary appliance that needs to be activated. <arg2>, etc, represent additional appliances that can be optionally activated simultaneously.',

'turn_off': 'turn_off(<arg1>) is used to deactivate various devices or utilities. <arg1>refers to the object or device to be deactivated, such as a 'socket', 'tap', or 'blending machine'.',

'cut': 'cut(<arg1>, <arg2>) is used to perform the action of cutting on various items. <arg1>refers to the item to be cut, which is mandatory. <arg2>is optional and denotes the tool used for cutting; if <arg2>is omitted, a standard cutting tool is assumed.',

'throw': 'throw(<arg1>, <arg2>) is used to dispose of or place an object in a specified or default location. <arg1>refers to the item to be disposed of or relocated, whereas <arg2>is optional and designates the location where the item should be placed. If <arg2>is omitted, the function selects a default disposal method or location based on the item or context.',

'mix': 'mix(<arg1>, <arg2>) is used to combine or stir ingredients, typically in a cooking context. <arg1>refers to the item or ingredients to be mixed, and <arg2>is optional and denotes the tool used for mixing, such as a spoon or paddle. When <arg2>is omitted, the method of mixing is unspecified or assumed to be manual.',

'touch': 'touch(<arg1>, <arg2>) is used to simulate the action of touching various items or materials. <arg1>refers to the object or material that is the primary focus of the action, whereas <arg2>is optional and provides additional detail on a specific part of the item to touch, assuming a generic aspect if omitted.',

'eat': 'eat(<arg1>, <arg2>) is used to perform the action of consuming a specified item. <arg1>refers to the item to be consumed. <arg2>is optional and describes the method by which the food is to be eaten, for example, 'slowly'.',

'pull': 'pull(<arg1>, <arg2>) is used to simulate the action of pulling something within a specific context. <arg1>refers to the object that is being pulled, such as a drawer or an oven grill. <arg2>is optional and describes a secondary reference or location, like a pan or a steel cabinet, which adds context to where the object is located or what it is associated with. If <arg2>is omitted, the action focuses solely on <arg1>.',

'unfold': 'unfold(<arg1>, <arg2>=None) is used to expand or open various types of items. <arg1>refers to the item to be unfolded, such as fabric, body parts, or food items. <arg2>is optional and allows for additional specifications on how the unfolding should be performed, tailored based on the nature of the item. If <arg2>is omitted, basic operations are performed.',

'dip': 'dip(<arg1>, <arg2>) is used to immerse an item into a container. <arg1>refers to the item to be dipped, such as 'dough' or 'hand', and <arg2>describes the container like 'bowl of water' or 'flour'. This function facilitates operations involving coating or soaking an item.',

'observe': 'observe(<arg1>) is used to examine the specified environment or objects. <arg1>refers to an array containing one or more strings that describe what should be focused on during the observation. At least one string is mandatory to define the scope of observation, while additional strings are optional to provide more detail.',

'taste': 'taste(<arg1>, <arg2>) is used to simulate the action of tasting a specified item with or without a utensil. <arg1>refers to what is being tasted, such as food or soup. <arg2>is optional and specifies the utensil used for tasting, like a spoon. If <arg2>is omitted, the action of tasting is assumed to be done without any specific utensil.',

'apply': 'apply(<arg1>, <arg2>) is used to perform operations involving the application or manipulation of cooking ingredients or tools. <arg1>refers to the primary material or tool being used, such as 'flour' or 'oil'. <arg2>is optional and typically refers to the target where <arg1>is applied, like 'dough' or 'frying pan'.',

'switch': 'switch(<arg1>) is used to change or replace the current tool in use within a system or application. <arg1>corresponds to the name of the tool that the function will switch to.',

'roll': 'roll(<arg1>, <arg2>) is used to flatten or shape an item using a tool. <arg1>refers to the item to be rolled, such as dough or foil. <arg2>is optional and indicates the tool used for rolling, like a 'rolling pin' or 'rolling board'. If <arg2>is not specified, a default tool or method is used to roll <arg1>.',

'lay': 'lay(<arg1>, <arg2>) is used to place objects or substances within a specific environment or a default setting if not specified. <arg1>refers to what is being placed, and <arg2>is optional and defines where the item is placed.',

'gesture': 'gesture(<arg1>, <arg2>, etc) is used to perform low-level actions based on the type of gesture or action specified. <arg1>is mandatory and refers to the string specifying the type of gesture or action to be executed. <arg2>is optional and allows for additional details or modifications to the gesture when necessary.',

'steer': 'steer(<arg1>, <arg2>, etc) is used to manipulate or interact with an object in a controlled environment. <arg1>refers to any object that requires handling or operation. <arg2>is optional, enhancing or specifying the nature of the interaction.',

'operate': 'operate(<arg1>, <arg2>) is used to activate or manage a specified device. <arg1>refers to the name of the device being operated, while <arg2>is optional and allows specific operational parameters to be passed, such as temperature, duration, or intensity.',

'store': 'store(<arg1>, <arg2>) is used to log or record items into a storage system. <arg1>refers to the list of items to be stored, which can include a single item or multiple items listed together. <arg2>is optional and specifies where the items are to be stored, indicating the physical or logical grouping.',

'tilt': 'tilt(<arg1>, <arg2>) is used to tip or angle an item, often to enable actions like pouring. <arg1>refers to the item that needs to be tilted. <arg2>is optional and defines the degree or direction of tilt. If <arg2>is omitted, a default tilt setting is used.',

'lift': 'lift(<arg1>, <arg2>) is used to simulate the action of picking up or lifting an object or a group of objects. <arg1>refers to the primary object to be lifted, and <arg2>is optional, indicating an additional item or tool used alongside the primary object during the lifting process.',

'scrape': 'scrape(<arg1>, <arg2>) is used to perform the action of scraping one item against another. <arg1>refers to the item to be scraped, which is mandatory, such as 'cabbage' or 'vegetables'. <arg2>is optional and refers to the surface or tool against which the item is scraped, like 'board' or 'frying pan'. If <arg2>is omitted, the function defaults to a generic, predefined scraping context.',

'bend': 'bend(<arg1>, <arg2>, <arg3>) is used to modify the shape or structure of an object.

*<arg1>refers to the object undergoing the bending. <arg2>and <arg3>are optional and specify the degree and the direction of the bend, respectively, allowing for precise control over the bending process.',*

*'hit': 'hit(<arg1>, <arg2>) is used to simulate the action of one object striking another. <arg1>refers to the primary object being hit, while <arg2>is optional and indicates any additional object used in the hitting action, such as a tool.',*

*'reduce_heat': 'reduce_heat(<arg1>, <arg2>) is used to lower the temperature or heat output of a specific device. <arg1>refers to the device on which the heat reduction is to be applied, and <arg2>is optional and provides an interface or method for achieving the heat reduction, allowing for precise control when specified.',*

*'rub': 'rub(<arg1>, <arg2>) is used to simulate the action of rubbing an object or surface. <arg1>refers to the primary object that is being rubbed, and <arg2>is optional, referring to a secondary object or surface involved in the rubbing, which can enhance or alter the rubbing context. If <arg2>is omitted, the rubbing action is considered to be performed solely with <arg1>.',*

*'add': 'add(<arg1>, <arg2>) is used to simulate placing an item into a container or context within a simulated environment. <arg1>refers to the object to be added, which is mandatory. <arg2>is optional and specifies the location or receptacle for the item. If <arg2>is omitted, the item is added to a default location or context.',*

*'mould': 'mould(<arg1>) is used to shape or form a material into a desired structure. <arg1>refers to the substance that needs to be shaped, such as clay, dough, or plastic.',*

*'knead': 'knead(<arg1>, <arg2>) is used to manipulate and prepare materials. <arg1>refers to the primary material to be kneaded, such as dough or clay. <arg2>is optional and denotes the surface or item against which the kneading is performed, like a tray or a rolling board.',*

*'stop': 'stop(<arg1>, <arg2>) is used to terminate an ongoing process. <arg1>refers to the type of process being stopped, such as 'liquid'. <arg2>is optional and specifies the equipment involved, like 'gas cooker'. If <arg2>is not provided, the function defaults to stopping all processes related to <arg1>.',*

*'cook': 'cook(<arg1>, <arg2>) is used to simulate the cooking process of a specified ingredient with or without a utensil. <arg1>refers to the item to be cooked, which is a mandatory argument. <arg2>is optional and specifies the tool used in the cooking process, defaulting to none if not provided.',*

*'rest': 'rest(<arg1>, <arg2>) is used to model the passive placement of one object against or on another. <arg1>refers to the primary object that is being supported or placed, while <arg2>is optional and refers to the object or surface against which <arg1>is resting. If <arg2>is omitted, the function defaults to a predetermined resting position or surface.',*

*'increase_temperature': 'increase_temperature(<arg1>, <arg2>) is used to raise the temperature of a device using a control mechanism. <arg1>refers to the device whose temperature needs to be increased, such as a cooker or heater. <arg2>is optional and refers to the specific method or interface, like a control knob or button, used to increase the temperature; if not specified, a default method is used.',*

*'dab': 'dab(<arg1>, <arg2>) is used to absorb or blot excess liquid or substances from items. <arg1>refers to the object that requires dabbing, while <arg2>is optional and specifies the material used for dabbing. If <arg2>is omitted, a standard method of dabbing is applied.',*

*'fix': 'fix(<arg1>, <arg2>) is used to attach or affix <arg1>to <arg2>. <arg1>refers to the object or component that needs to be fixed, while <arg2>is optional and identifies the target object or location to which <arg1>will be attached. If <arg2>is omitted, <arg1>is attached to a default object or location.',*

*'dry': 'dry(<arg1>, <arg2>) is used to remove moisture from specified items. <arg1>refers to the item needing drying, like "hands" or "mango." <arg2>is optional and indicates the material used to aid the drying, such as "towel" or "napkin."',*

*'hang': 'hang(<arg1>, <arg2>) is used to place an object onto a specified or default location for storage or accessibility. <arg1>refers to the object to be hung, and <arg2>is optional and denotes the location where the object should be placed. If <arg2>is omitted, a default location is used.',*

*'tie': 'tie(<arg1>, <arg2>) is used to wrap or secure items. <arg1>refers to the material used for tying, such as strings or wraps. <arg2>is optional and indicates additional materials or conditions that might affect the tying process, such as environmental factors or secondary materials.',*

*'sprinkle': 'sprinkle(<arg1>, <arg2>) is used to apply a substance over a surface or object. <arg1>refers to the material to be sprinkled, which is mandatory. <arg2>is optional and de-*

*fines the surface or object where <arg1>is to be applied. If <arg2>is omitted, the substance is applied to a default location.',*

*'swing': 'swing(<arg1>) is used to alter or move an object in a predefined manner. <arg1>refers to the object being manipulated and the specific actions depend on the nature of this object.',*

*'fill': 'fill(<arg1>, <arg2>) is used to insert a specified substance into a designated container. <arg1>refers to the container that will contain the substance, and <arg2>describes the substance to be filled into the container.',*

*'wear': 'wear(<arg1>, <arg2>, <arg3>) is used to simulate the action of dressing a character or entity with a specific item. <arg1>is mandatory and refers to the item to be worn, described as a string. <arg2>and <arg3>are optional, allowing for customization of style and size, respectively.',*

*'unsure': 'unsure(<arg1>, <arg2>, etc) is used to perform an ambiguous action based on the provided context or data. <arg1>is a mandatory parameter that provides the necessary context or data for the operation of the function. <arg2>and other additional arguments are optional and enhance the function's flexibility and adaptability to varying use cases.',*

*'sort': 'sort(<arg1>, <arg2>) is used to organize or prioritize items based on specific criteria. <arg1>is mandatory and specifies the operation to be performed, while <arg2>is optional and includes the items to be sorted. This function can be used with varying numbers of arguments to adapt to different sorting requirements or settings.',*

*'stretch': 'stretch(<arg1>) is used to modify the physical state of a malleable material by elongating or thinning it. <arg1>refers to the malleable material that is altered by the function.',*

*'squeeze': 'squeeze(<arg1>, <arg2>, etc) is used to compress or reduce the size of various types of input objects. <arg1>refers to the object or substance to be compressed. <arg2>and other optional arguments can be added to modify the function based on the specifics of the compression or the context in which it is applied.',*

*'flatten': 'flatten(<arg1>, <arg2>) is used to press and spread a material into a flatter shape. <arg1>is mandatory and specifies the material to be flattened, while <arg2>is optional and represents a tool used to assist in the flattening process. This function is generally used when a uniform thickness is desired or to prepare the material for further processing.',*

*'climb': 'climb(<arg1>) is used to simulate or command an entity to ascend or mount a specified target. <arg1>refers to the object or location that the entity should climb onto.',*

*'interact': 'interact(<arg1>, <arg2>) is used to perform interactions with various entities or objects. <arg1>refers to the entity or object to interact with, which is mandatory. <arg2>is optional and specifies the method or type of interaction desired; if omitted, it defaults to a standard interaction mode.'}*

[end of actionable function and documentation]

*Based on the low-level actionable actions provided, you will need to make one or more function calls in order to achieve the mid-level step described in the question.*

*Respond needs to strictly be a list of these actionable functions following this format: "fuction1(args)","fuction2(args)","fuction3(args)", ...*

*Besides these functions, your response should not contain anything else,these functions should not be numbered or explained, simply separated by commas and output directly.*

*For example: "put(jar, cabinet)","grab(jar)","mix(jar)","put(jar, cabinet)".*

*You should not include any other text in your response.*

*Question: {question}*

*List of actionable functions:*

- **Object grounding in visual grounding:** {*question*} *Please give out the bounding box coordinates of the object.*

- **Frame grounding in visual grounding:** {*question*} *Analyze the provided video and identify the frame number of the last keyframe that is relevant to the specified query. Please provide only the frame number as your response.*

- **Temporal grounding in visual grounding:** {*question*} *Please provide the starting and ending times for that step.*

- **Critique in reward modeling:** *Imagine you are the camera wearer (I) who recorded the video. Please directly answer yes or no to determin whether the task is completed or not. Question: {question} Short answer:*

- **Feedback in reward modeling:** *Imagine you are the camera wearer (I) who recorded the video. The video contains an uncompleted task. Please identify the essential completion signals in my*

*observations that indicate the task is not completed by me. Please directly generate the rationale as short as possible. Question: {question} Short Answer:*

### D.2 EVALUATION PROMPTS

Here we list the prompts for API-based models to assess the performance for some tasks.

- **Video question answering:** *[Instruction]\nPlease act as an impartial judge and evaluate the quality of the response provided by an AI assistant to the user question displayed below. Your evaluation should consider correctness and helpfulness. You will be given a reference answer and the assistant's answer. Begin your evaluation by comparing the assistant's answer with the reference answer. Identify and correct any mistakes. The assistant has access to an image alongwith questions but you will not be given images. Therefore, please consider only how the answer is close to the reference answer. If the assistant's answer is not exactly same as or similar to the answer, then he must be wrong. Be as objective as possible. Discourage uninformative answers. Also, equally treat short and long answers and focus on the correctness of answers. After providing your explanation, you must rate the response with either 0, 0.5 or 1 by strictly following this format: "[[rating]]", for example: "Rating: [[0.5]]".\n\n[Question]\n{question}\n\n[The Start of Reference Answer]\n{ref_answer_1}\n[The End of Reference Answer]\n\n[The Start of Assistant's Answer]\n{answer}\n[The End of Assistant's Answer]"*

- **High-to-mid in hierarchy planning:** *[Instruction]\nPlease act as an impartial judge and evaluate the quality of the response provided by an AI assistant to the user question displayed below. Your evaluation should consider correctness and helpfulness. You will be given a reference answer and the assistant's answer. Begin your evaluation by comparing the assistant's answer with the reference answer. Identify and correct any mistakes. The assistant has access to an image alongwith questions but you will not be given images. Therefore, please consider only how the answer is close to the reference answer. The reference answer and the assistant's answer both describe a mid-level step towards completing a high-level goal, you must consider if these two mid-level steps are similar. If the assistant's answer is not exactly same as or similar to the answer, then he must be wrong. Be as objective as possible. Discourage uninformative answers. Also, equally treat short and long answers and focus on the correctness of answers. After providing your explanation, you must rate the response with either 0, 0.5 or 1 by strictly following this format: "[[rating]]", for example: "Rating: [[0.5]]".\n\n[Question]\n{question}\n\n[The Start of Reference Answer]\n{ref_answer_1}\n[The End of Reference Answer]\n\n[The Start of Assistant's Answer]\n{answer}\n[The End of Assistant's Answer]*

- **Mid-to-low in hierarchy planning:** *[Instruction]\nPlease act as an impartial judge and evaluate the quality of the response provided by an AI assistant to the user question displayed below. Your evaluation should consider correctness and helpfulness. You will be given a reference answer and the assistant's answer. Begin your evaluation by comparing the assistant's answer with the reference answer. Identify and correct any mistakes. The assistant has access to an image alongwith questions but you will not be given images. Therefore, please consider only how the answer is close to the reference answer. The reference answer and the assistant's answer both describe a trajectory of low-level automic actions towards completing a mid-level step, you must consider if these two trajectories of low-level atomic actions are similar, especially the key actions to achieve the mid-level step. If the assistant's answer is not exactly same as or similar to the answer, then he must be wrong. Be as objective as possible. After providing your explanation, you must rate the response on a scale of 0 to 10 by strictly following this format: "[[rating]]", for example: "Rating: [[5]]".\n\n[Question]\n{question}\n\n[The Start of Reference Answer]\n{ref_answer_1}\n[The End of Reference Answer]\n\n[The Start of Assistant's Answer]\n{answer}\n[The End of Assistant's Answer]*

- **Feedback in reward modeling:** *[Instruction]\nPlease act as an impartial judge and evaluate the quality of the response provided by an AI assistant to the user question displayed below. Your evaluation should consider correctness and helpfulness. You will be given three reference answers and the assistant's answer. Begin your evaluation by comparing the assistant's answer with the reference answers. Identify and correct any mistakes. The assistant has access to an image alongwith questions but you will not be given images. Therefore, please consider only how the answer is close to the reference answers. If the assistant's answer is not exactly same as or similar to all reference answers, then he must be wrong. If the assistant's answer is exactly same as or similar to*

*any one reference answer, then it is correct. Be as objective as possible. Discourage uninformative answers. Also, equally treat short and long answers and focus on the correctness of answers. After providing your explanation, you must rate the response with either 0, 0.5 or 1 by strictly following this format: "[[rating]]", for example: "Rating: [[0.5]]".\n\n[Question]\n{question}\n\n[The Start of Reference Answer]\n{ref_answer_1}\n[The End of Reference Answer]\n\n[The Start of Assistant's Answer]\n{answer}\n[The End of Assistant's Answer]*

# E  CASE STUDIES

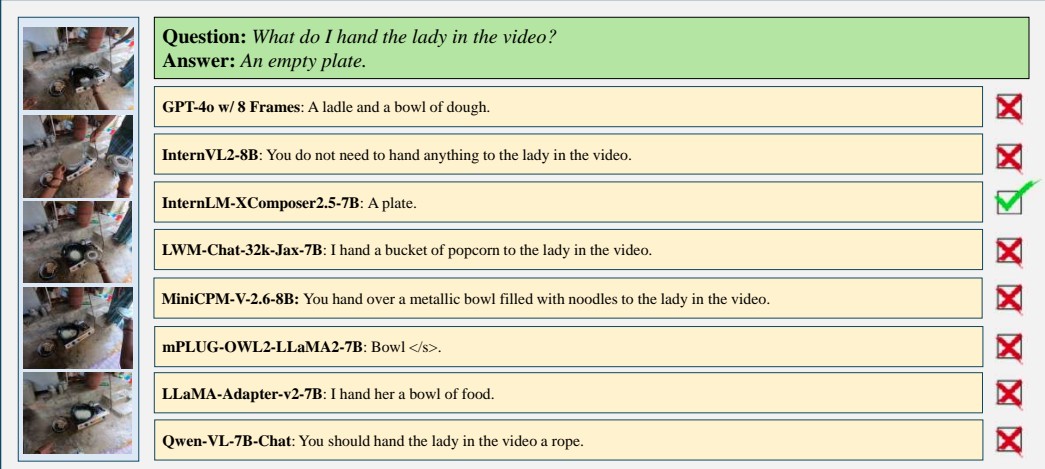

Figure 6: Case of object interaction in video question answering.

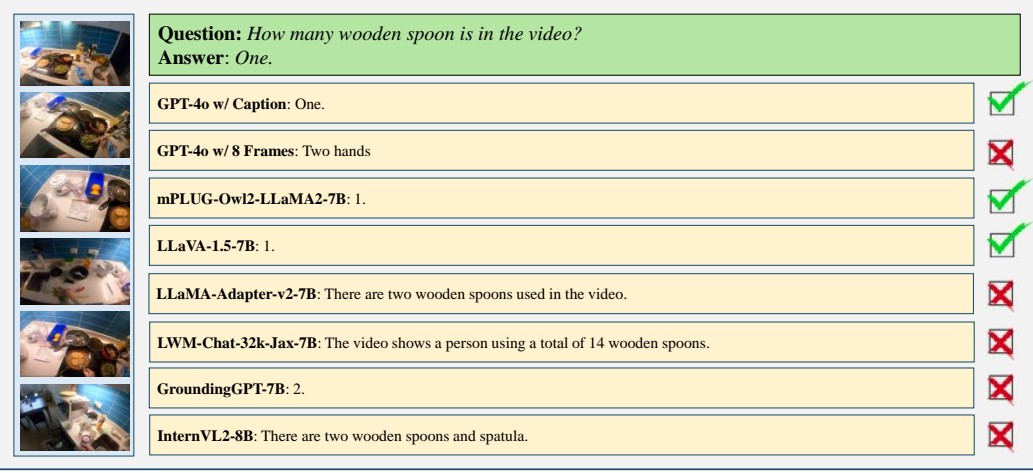

Figure 7: Case of object count of video question answering.

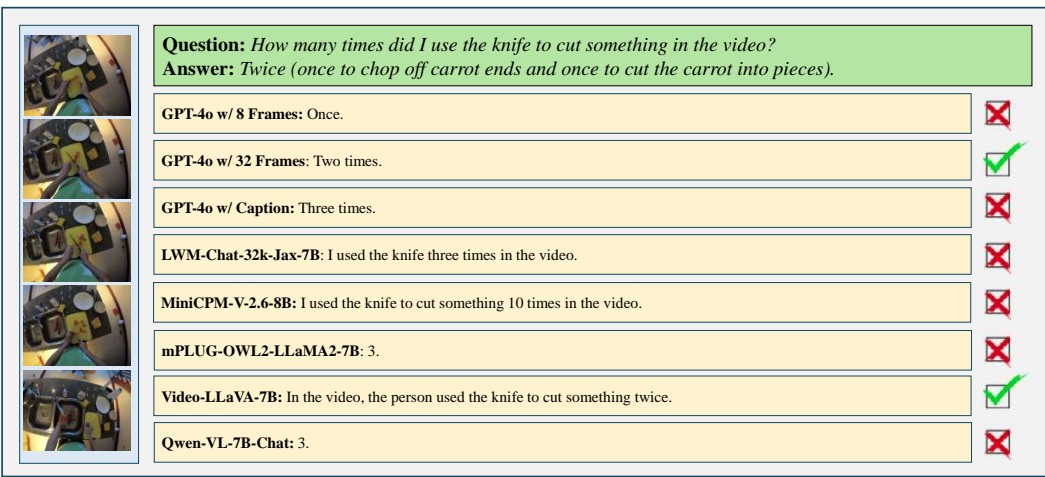

Figure 8: Case of action count of video question answering.

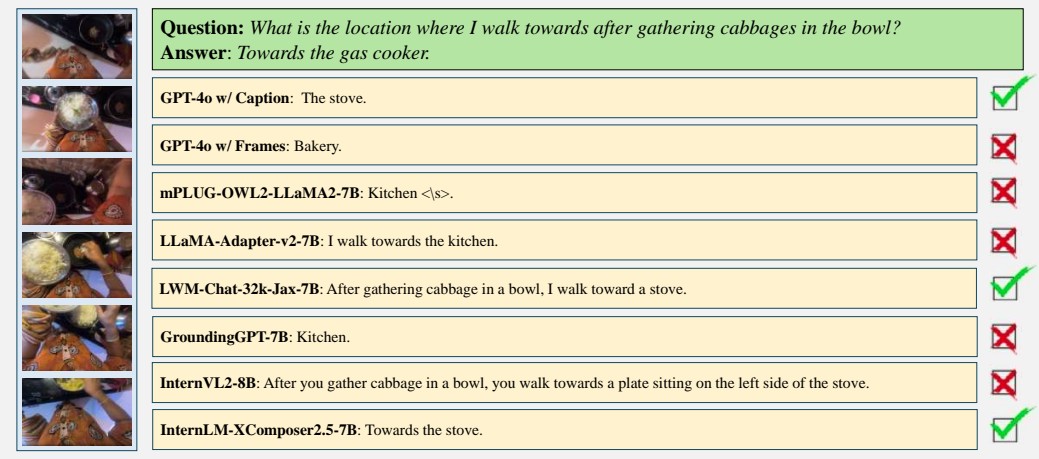

Figure 9: Case of scene transition of video question answering.

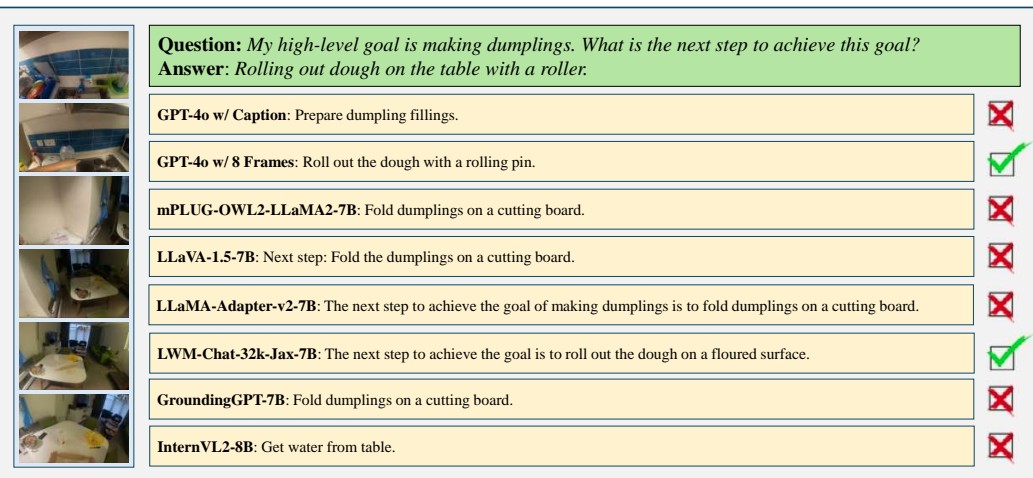

Figure 10: Case of the high-to-mid task in hierarchy planning.

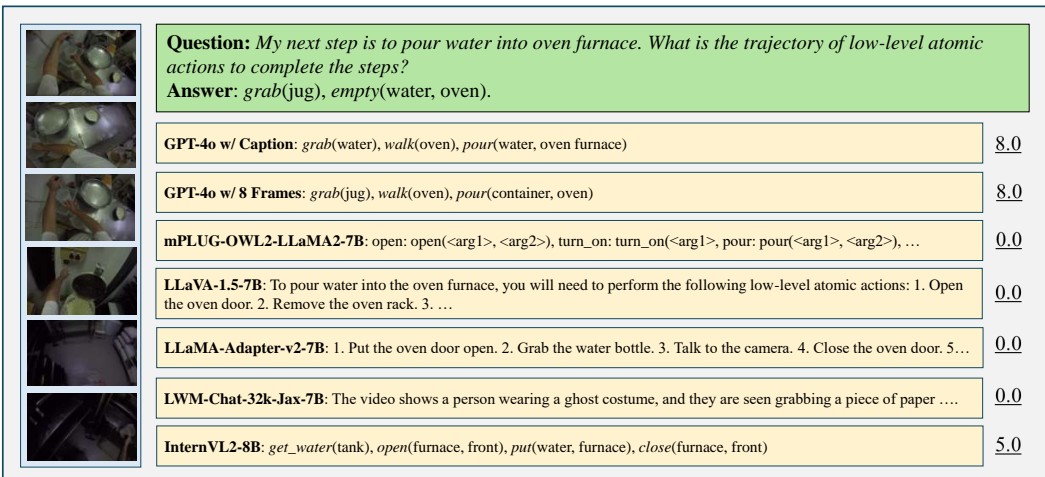

Figure 11: Case of the mid-to-low task in hierarchy planning.

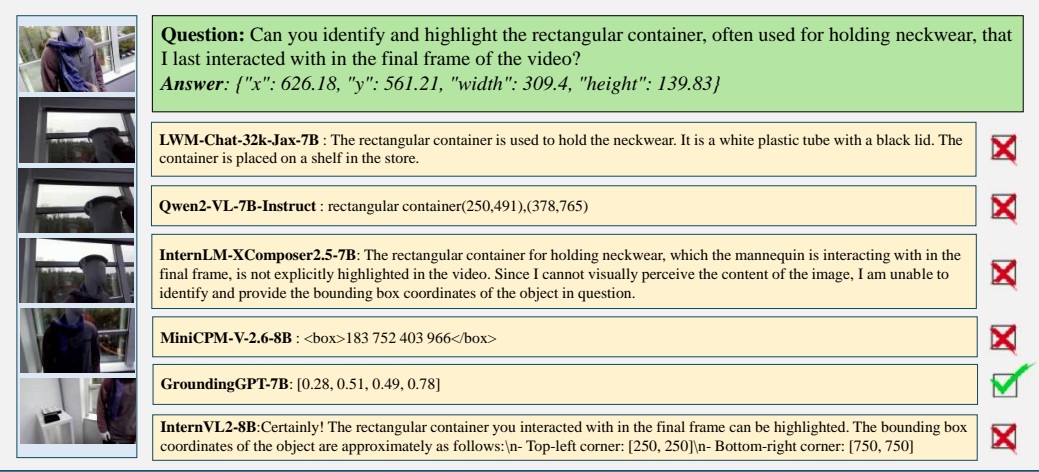

Figure 12: Case of the object grounding in visual grounding. The output of GroundingGPT represents percentage.

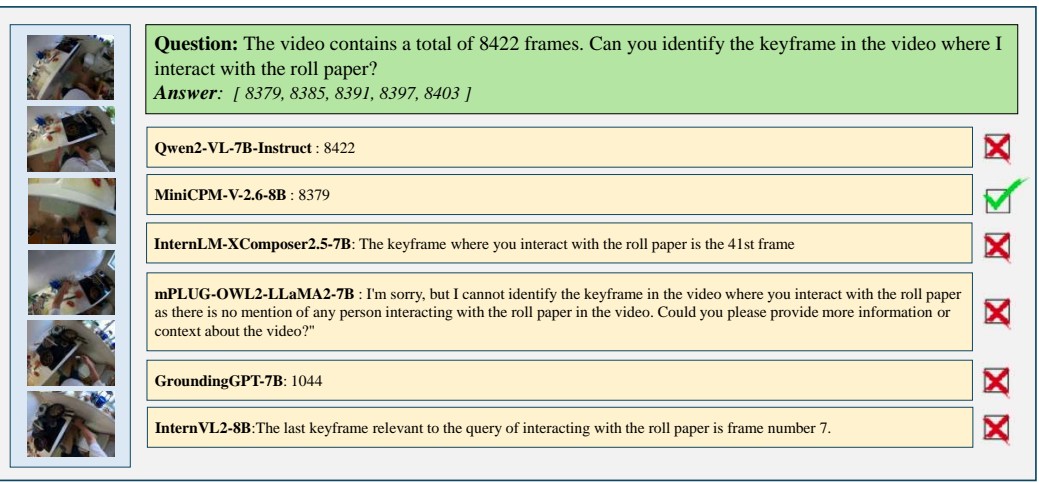

Figure 13: Case of frame grounding in visual grounding.

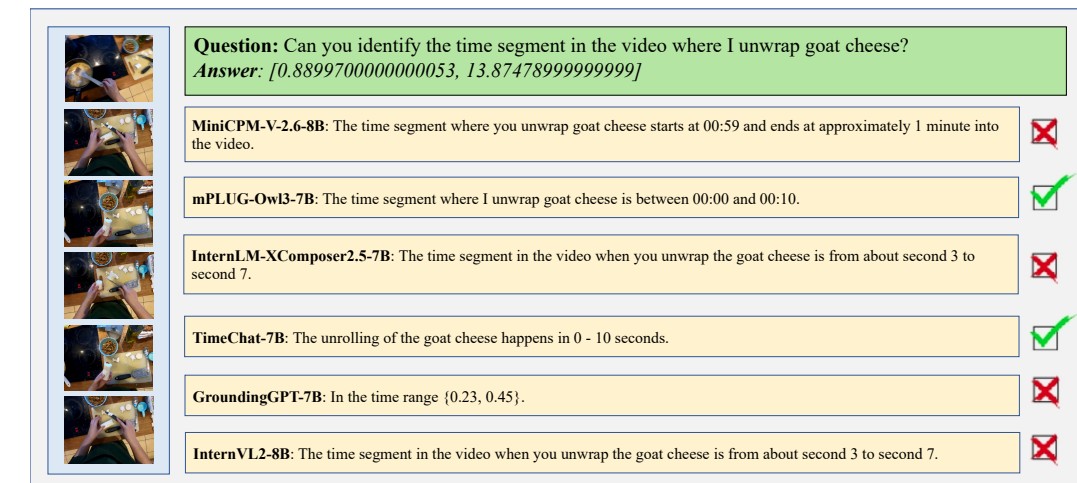

Figure 14: Case of temporal grounding in visual grounding.

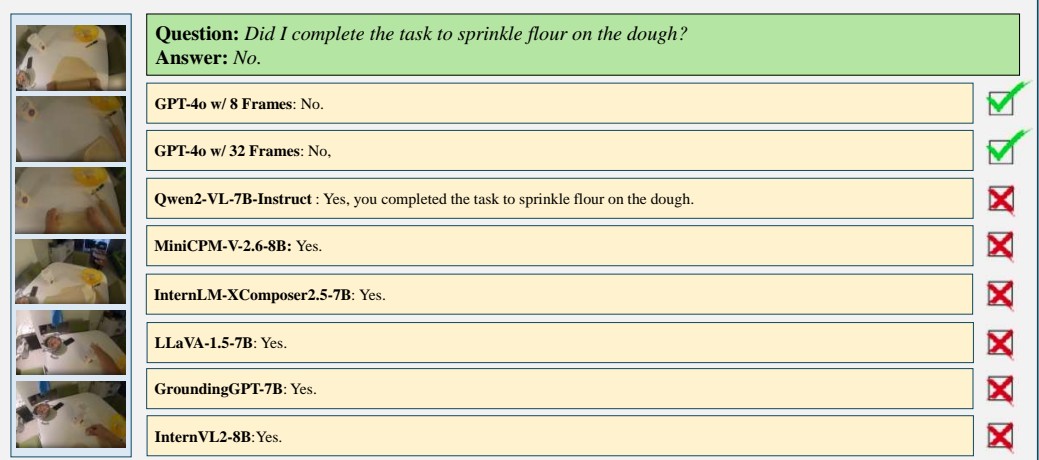

Figure 15: Case of the critique task in reward modeling.

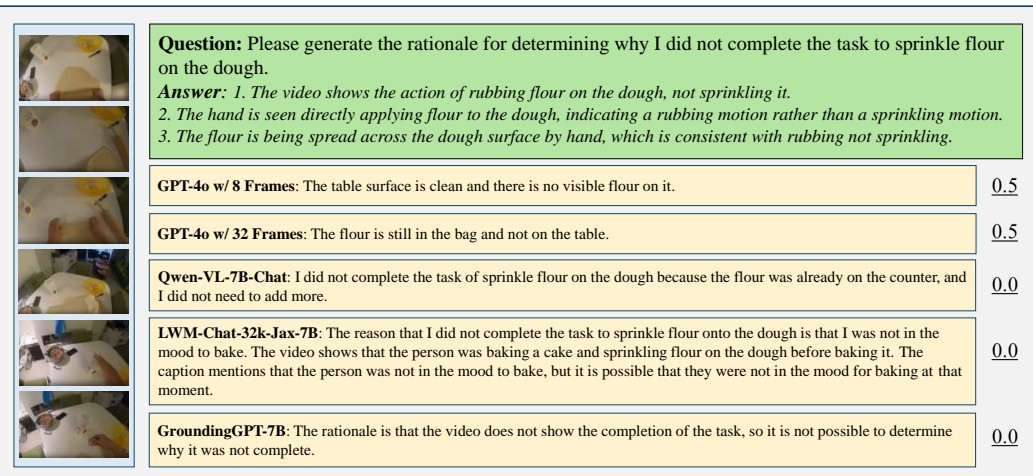

Figure 16: Case of the feedback task in reward modeling.

