# OpenReview forum: "VidEgoThink: Assessing Egocentric Video Understanding Capabilities for Embodied AI"
_ICLR.cc/2025/Conference — ICLR 2025 Conference Withdrawn Submission_

### Official Review · Reviewer_ptWK · 2024-10-18

**Soundness:** 3
**Presentation:** 3
**Contribution:** 2
**Rating:** 3
**Confidence:** 5

**Summary:**

The paper introduces VidEgoThink, a benchmark for evaluating egocentric video understanding capabilities of Multi-modal Large Language Models (MLLMs). It has four key tasks: video question-answering, hierarchy planning, visual grounding, and reward modeling. The authors develop an automatic data generation pipeline to generate the data using GPT-4o from Ego4D dataset. Human annotators are employed to filter the generated data. Experiments are conducted with various MLLMs, including API-based, open-source image-based, and video-based models, showing that current models are still lacking in egocentric video understanding.

**Strengths:**

1. The benchmark is comprehensive. Four key tasks of video question-answering, hierarchy planning, visual grounding, and reward modeling are designed, and many models including open and closed source LLMs are evaluated.

2. The experimental results reveal that even the most advanced MLLMs struggle with egocentric video understanding, providing valuable insights into areas for potential research and development.

**Weaknesses:**

1. The data generation pipeline heavily relies on GPT-4o. The authors claim to use manual inspection on the automatically generated data, however, no detail on the human filtering is provided. For example, how the inspection is done, what instructions are given to the annotators, how many percentages of data are filtered, where the annotators are hired, and at what salary.

2. The benchmark is mainly generated using GPT-4o, and the evaluation is in an open-ended manner. The evaluator is also GPT-4o. This raises the concern that the evaluation is not fair to other models. The evaluation may be not entirely based on the correctness, but partially based on the (language style) similarity to GPT-4o.

3. For the mid-to-low planning task, the authors converted the output style into a "function-like" format. Since the function here is purely verb and noun pairs, I think this is totally unnecessary. It can be also seen from the example figure of this task, where not all models can follow the instructions to generate the answers of the same format, which potentially influences the evaluation result. From the result, all models except GPT-4o get almost completely wrong outputs. I understand the author's claim that this is closer to the "actions" of embodied AI, however in this work it is a simple text re-writing, and I think it is more harmful to the evaluation compared with being "closer to embodied AI".

4. For the object grounding task, the prompt given to the LLMs does not seem to contain enough information. The evaluation is on the last frame of the video, but the prompt just says "Please give out the bounding box coordinates of the object." (L1231)

5. The title of this paper is "for embodied AI". However, only real-world human data is used. While it is understandable that the design of tasks is inspired by embodied AI, I believe this claim is not appropriate. This benchmark is more on the egocentric video understanding side.

**Questions:**

It would be helpful if the authors could clarify the human annotator filtering process and the explanation for weakness points 2,3,4.

---

### Official Review · Reviewer_qHFD · 2024-10-30

**Soundness:** 2
**Presentation:** 3
**Contribution:** 2
**Rating:** 6
**Confidence:** 4

**Summary:**

The paper introduces VidEgoThink, a benchmark for evaluating egocentric video understanding in low-level control tasks within Embodied AI. VidEgoThink employs GPT-4o and prompt engineering to generate question-answer pairs from the Ego4D dataset, focusing on four tasks: video question-answering, hierarchical planning, visual grounding, and reward modeling. Human filtering is applied to ensure the quality of the benchmark. In subsequent evaluations, the benchmark reveals that all current multimodal large language models (MLLMs), including GPT-4o, perform poorly across these tasks.

**Strengths:**

The paper has several strong aspects:

1.This paper is the first to integrate four key tasks—video question-answering, hierarchical planning, visual grounding, and reward modeling—covering a broad spectrum from task planning and video understanding to spatial-temporal reasoning and fine-grained task completion awareness. This reflects the authors' deep understanding of the requirements for low-level control tasks in Embodied AI scenarios.

2.For each task, the authors attempt to define clear evaluation dimensions and metrics, providing a usable approach to assessing performance.

3.The experiments show that VidEgoThink is a robust benchmark that effectively highlights the limitations of current MLLMs in processing first-person perspective data.

**Weaknesses:**

I see the following as weak points:
1. The quality of VidEgoThink's benchmark data is not thoroughly analyzed.

The paper employs GPT-4o and prompt engineering to generate question-answer pairs from the Ego4D dataset. However, the quality of the benchmark data is not thoroughly analyzed. While it is mentioned that 3 human filters were applied, the filtering criteria are not clearly explained. The statistical analysis in the appendix only covers aspects such as question and answer length, and scene types, which are likely based on the Ego4D dataset's categories. These analyses provide little insight into the overall quality of the benchmark.

I would have liked to see more detailed insights, such as which tasks and actions are included in the benchmark, information on long-term and mid-term tasks along with their difficulty,  as well as which scenes are more challenging, and which are easier. The varying video lengths, differences in activity density, and changing camera perspectives in Ego4D are factors that likely contribute to significant difficulties in video understanding, but these are not addressed.


2. The excessive length of egocentric videos, combined with the inherent difficulty in understanding egocentric scenes, likely leads to the performance decline of all current MLLMs.

Table 3 of the paper shows that the average length of videos in VidEgoThink is 270.74 seconds. However, in the experimental evaluation, only 32 frames, 8 frames, or even captions are used as input. Given that Ego4D has a frame rate of 30 fps and an average video length of around 270 seconds, selecting 32 frames results in sampling roughly one frame every 250 frames. This sparse sampling from such long videos significantly hampers the video perception and understanding capabilities of current MLLMs.

For egocentric videos with complex scenes, numerous background objects, rapid camera movements, and transitions between scenes, such limited input is unlikely to provide sufficient information about the video. This likely also contributes to the generally poor performance of MLLMs.

**Questions:**

Refer to weaknesses.

---

### Official Review · Reviewer_mQwi · 2024-10-30

**Soundness:** 1
**Presentation:** 2
**Contribution:** 1
**Rating:** 3
**Confidence:** 3

**Summary:**

This paper introduces a new benchmark to evaluate Multimodal Large Language Models (MLLMs) for egocentric video understanding, advancing their applications in Embodied AI. The benchmark includes four tasks: video QA, hierarchical planning, visual grounding, and reward modeling. Annotations are derived from the Ego4D dataset, processed by GPT-4o, and refined by human annotators to meet task requirements. Experiments with state-of-the-art image and video MLLMs reveal significant gaps in specific competencies.

**Strengths:**

1. Developing benchmarks to advance MLLMs in Embodied AI remains relatively under-explored and warrants deeper investigation.

2. The proposed benchmark, VidEgoThink, encompasses diverse tasks to challenge MLLMs across multiple dimensions.

3. Experimental results highlight the limitations of current MLLMs in egocentric video understanding, pointing to important directions for future research.

**Weaknesses:**

**1. Limited Contributions**

1.1 The proposed benchmark primarily combines tasks from prior work, such as egocentric video question-answering [1,2] and hierarchical planning [3,4], with data drawn from established annotations—Ego4D narrations for Video QA, goal-step hierarchies for planning, and Visual Queries for grounding. Moreover, similar studies have leveraged LLMs to process Ego4D annotations [2,5,6]. Could the authors clarify how VidEgoThink's approach to processing Ego4D annotations distinguishes itself from or advances existing works? Table 3 lacks persuasive evidence, as previously noted, the ‘task types’ predominantly stem from existing benchmarks.

1.2 Despite noting poor performance in planning and grounding tasks, the authors neither suggest specific enhancements nor discuss pathways for future model improvement. What targeted model enhancements do the authors suggest to address these shortcomings? Additionally, how might advancements on this benchmark translate into practical applications for embodied AI? I encourage the authors to provide a detailed discussion on these points.

**2. Ambiguities in Design and Conclusions**

2.1 The stated motivation is to advance embodied AI (L53), yet the evaluation approach—presenting a complete video and question to the MLLMs—deviates from typical embodied agent operations, which would ideally process video in a streaming fashion [7,8]. Could the authors discuss how their evaluation method aligns with or differs from real-world embodied AI applications?

2.2 The rationale behind defining the dimensions of Video QA is unclear. For example, why is there an object prediction but no action prediction?

2.3 The necessity of visual input for the Mid-to-Low Planning task is unclear. Based on the task description and Figure 3, this task appears potentially solvable using LLM knowledge alone. Performance results in Table 2 show that GPT-4o performs better with 8 frames than with 32 frames, which may support this view. However, the lower performance of the ‘only-qa’ and ‘captions’ baselines suggests inconsistencies. Could the authors clarify why visual input is necessary for the Mid-to-Low Planning task and explain the seemingly contradictory results?

2.4 MSE is listed as the metric for frame grounding (L300), yet it contradicts Equation 3. Is this a typo?

2.5 While the authors mention that GPT-4o does not support video input (L425), they do not explore API models such as Gemini 1.5, which do have video input capabilities. This omission raises questions about the choice of model for evaluation.

---
[1] Majumdar, Arjun, et al. "Openeqa: Embodied question answering in the era of foundation models." In CVPR. 2024.

[2] Di, Shangzhe, and Weidi Xie. "Grounded Question-Answering in Long Egocentric Videos." In CVPR. 2024.

[3] Shridhar, Mohit, et al. "Alfred: A benchmark for interpreting grounded instructions for everyday tasks." In CVPR. 2020.

[4] Song, Yale, et al. "Ego4d goal-step: Toward hierarchical understanding of procedural activities." In NeurIPS. 2024.

[5] Ramakrishnan, Santhosh Kumar, Ziad Al-Halah, and Kristen Grauman. "Naq: Leveraging narrations as queries to supervise episodic memory." In CVPR. 2023.

[6] Mangalam, Karttikeya, Raiymbek Akshulakov, and Jitendra Malik. "Egoschema: A diagnostic benchmark for very long-form video language understanding." In NeurIPS. 2023.

[7] Qian, Rui, et al. "Streaming long video understanding with large language models." arXiv:2405.16009 (2024).

[8] Zhang, Haoji, et al. "Flash-VStream: Memory-Based Real-Time Understanding for Long Video Streams." arXiv:2406.08085 (2024).

**Questions:**

My questions have been included in the Weaknesses section. I primarily suggest that the authors provide additional explanations and analysis.

---

### Official Review · Reviewer_TXuC · 2024-11-03

**Soundness:** 3
**Presentation:** 4
**Contribution:** 2
**Rating:** 5
**Confidence:** 4

**Summary:**

This paper presents benchmarks designed to assess the ability of Multi-modal Large Language Models to understand egocentric video, particularly for applications in Embodied AI. There are 4 benchmarks, Video Question Answering, Hierarchy Planning, Visual Grounding, and Reward Modeling, which are well aligned with Embodied AI applications.  The benchmark dataset is constructed using an automated pipeline that leverages the Ego4D dataset and GPT-4o to generate data, and there are 3 human inspectors. Extensive experiments are conducted with 14 MLLMs, including API-based models like GPT-4o, and open-source image-based and video-based models.

**Strengths:**

1. This paper is well-organized and clearly written.
2. The benchmark tasks are thoughtfully designed and effectively aligned with real-world Embodied AI applications.
3. The benchmark efficiently leverages the large-scale Ego4D dataset along with its existing annotations, and utilizes GPT to generate new, sensible labels.

**Weaknesses:**

1. Limited Insight from Experimental Results: Although the paper dedicates considerable space to clearly outlining the benchmark tasks in detail, the experimental analysis section is confined to just one page, making the overall work feel incomplete. The key takeaway from the analysis of this paper is that "MLLMs perform poorly across all tasks", but there is a lack of specific insights. For instance, the paper does not provide detailed analysis of performance variations under different test conditions or highlight which model structures perform better on certain tasks. Despite evaluating over 10 different MLLMs, the analysis focuses mainly on GPT-4o, neglecting a deeper exploration of the comparative performance of other models. Furthermore, while the paper emphasizes embodied AI applications, the experimental analysis does not offer actionable insights or future directions for MLLM design from an embodied AI perspective.

2. Limited Novelty in Benchmark Design: The Video Question Answering benchmark overlaps with existing egocentric video QA frameworks, such as EgoTaskQA: Understanding Human Tasks in Egocentric Videos. In Visual Grounding, the tasks like object detection, frame and temporal grounding adhere to traditional task formats, offering little innovation in task design. This reduces the impact of the benchmark’s contribution in terms of novelty.

**Questions:**

A potential reason for rejection could be that readers might perceive the near-zero accuracy of most models on the Hierarchy Planning and Visual Grounding tasks as a sign of flawed benchmark design or errors in the evaluation method. This perception could fundamentally undermine the paper’s contribution. Could you provide more details to explain this phenomenon? whether they align with expectations given the difficulty of the tasks?

---

### Official Review · Reviewer_6Z7v · 2024-11-03

**Soundness:** 3
**Presentation:** 3
**Contribution:** 3
**Rating:** 5
**Confidence:** 4

**Summary:**

The paper proposes a benchmark for evaluating Multimodal Large Language Models' capability to understand egocentric videos motivated by embodied AI. The proposed dataset, VidEgoThink, is built upon Ego4D and benchmarks a range of tasks, including video question answering, hierarchy planning, visual grounding, and reward modeling. The experiments evaluate GPT-4o and image/video-based MLLM methods. The findings suggest that foundation models still require significant advancements to be effectively applied to first-person scenarios in Embodied AI.

**Strengths:**

- MLLMs for embodied AI is a timely topic. The paper built the benchmark on Ego4D, which contains real-world recorded egocentric videos, providing a realistic environment for evaluating embodied AIs.

- Many benchmark designs reflect the paper's consideration of leveraging video content in embodied AI, e.g., separating VQA questions into object, action, and scene, last-frame-based object grounding, and reward modeling.

**Weaknesses:**

- My biggest concern is the data diversity. The authors also admitted that in the paper's limitations section. Many of the benchmark tasks only cover fewer than 50 original videos in Ego4D, while the entire Ego4D dataset contains 9,611 videos. As a concrete example, the Object subtask in VQA covers 29 original Ego4D videos spanning over 9 scenes. This yields to 3 videos per scene. Another example is the Object subtask in Visual Grounding, which yields fewer than 2 videos per scene for 25 scenes. It might not be a significant issue to reach the findings of the paper that all MLLMs suck. However, it is quite problematic for it to be potentially adapted by the research community for wider research. Hence, the usefulness of VidEgoThink in its current scale is limited.

- The paper sets good motivation for embodied AI, but in the real annotation, they seem to not be implemented. In the Introduction, the paper says that "compared to the absolute position in the whole environment (e.g., the microwave is in the kitchen), the relative position to Ego is more important (e.g., the microwave is one meter to my right) for interaction and manipulation." However, in the Figure 2 Scene Existence example, the question is "Where is the oven located in the video?" and the answer is "The oven is located in the kitchen." I didn't find other annotation examples that could match the Introduction claim.

- Ego4D videos are pre-recorded. Hence, they might have difficulty (or are even unsuitable to) evaluating task planning compared to an interactive environment. For example, to reach a high-level goal, there might be different orders of mid-level steps or omitting/adding mid-level steps as long as they lead to the final correct results. However, pre-recorded videos contain just one path for implementing the goal. This leads to a quite biased evaluation, especially when the number of videos covering the task is small, as in the proposed VidEgoThink.

- The paper uses GPT-4o to generate ground truth, e.g., feedback. The paper then evaluates GPT-4o. And, it is possible that the paper uses GPT-4o to implement method evaluation, e.g., g(\cdot) in Eq. 1. (I couldn't find details.) This seems problematic.

**Questions:**

Please address my concerns in the Weaknesses part.

---

### Note · Authors · 2024-12-03

I have read and agree with the venue's withdrawal policy on behalf of myself and my co-authors.